# Long Short-Term Imputer:
# Handling Consecutive Missing Values in Time Series

**Jiacheng You**                                                    *jiachengyou2hit@gmail.com*
*School of Computer Science and Technology, Harbin Institute of Technology (Shenzhen)*

**Xinyang Chen***                                                   *chenxinyang95@gmail.com*
*School of Computer Science and Technology, Harbin Institute of Technology (Shenzhen)*

**Yu Sun***                                                         *sunyu@nankai.edu.cn*
*College of Computer Science, DISSec, Nankai University*

**Weili Guan**                                                      *guanweili@hit.edu.cn*
*School of Computer Science and Technology, Harbin Institute of Technology (Shenzhen)*

**Liqiang Nie**                                                     *nieliqiang@gmail.com*
*School of Computer Science and Technology, Harbin Institute of Technology (Shenzhen)*

**Reviewed on OpenReview:** *https://openreview.net/forum?id=9NVJ0ZgEfT*

## Abstract

Encountered frequently in time series data, missing values can significantly impede time-series analysis. With the progression of deep learning, advanced imputation models delve into the temporal dependencies inherent in time series data, showcasing remarkable performance. This positions them as intuitive selections for time series imputation tasks which assume "Miss Completely at Random". Nonetheless, long-interval consecutive missing values may obstruct the model's ability to grasp long-term temporal dependencies, consequently hampering the efficacy of imputation performance. To tackle this challenge, we propose Long Short-term Imputer (LSTI) to impute consecutive missing values with different length of intervals. Long-term Imputer is designed using the idea of bi-directional autoregression. A forward prediction model and a backward prediction model are trained with a consistency regularization, which is designed to capture long-time dependency and can adapt to long-interval consecutive missing values. Short-term Imputer is designed to capture short-time dependency and can impute the short-interval consecutive missing values effectively. A meta-weighting network is then proposed to take advantage of the strengths of two imputers. As a result, LSTI can impute consecutive missing values with different intervals effectively. Experiments demonstrate that our approach, on average, reduces the error by 57.4% compared to state-of-the-art deep models across five datasets.

## 1 Introduction

Multivariate time series are abundant in real-world applications, including meteorology, finance, engineering, science, and healthcare (Hajihashemi & Popescu, 2015; Prakosa et al., 2012; Gupta et al., 2020; Tascikaraoglu et al., 2016; Zheng et al., 2015). These time series data often contain missing values due to reasons such as sensor failures and communication losses (Silva et al., 2012; Yi et al., 2016). These missing values impair the integrity and interpretability of the data, posing significant challenges to time series analysis (Hasan et al., 2021). Therefore, effectively imputing missing data before analysis is essential.

---

*Corresponding authors.

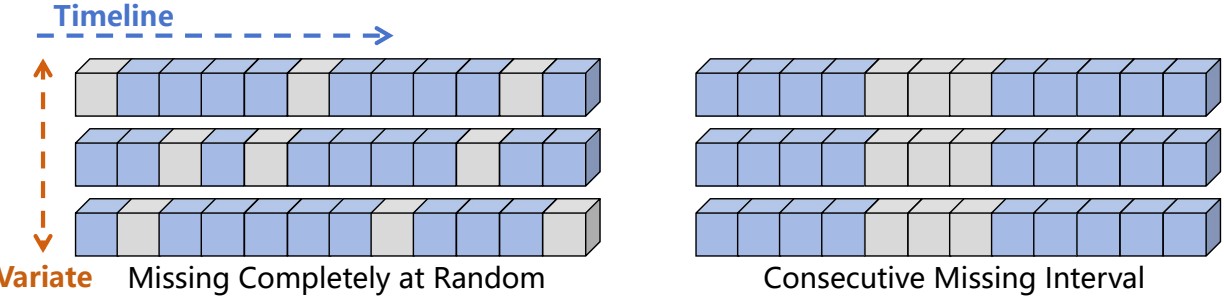

Figure 1: The difference between MCAR and long-interval consecutive missing values.

Many studies have addressed the task of imputing missing values using deep learning techniques (Wu et al., 2022; Du et al., 2023; Tashiro et al., 2021), and most of them are designed for scenarios involving Missing Completely at Random (MCAR). However, in the real world, missing values often occur consecutively and in clusters due to factors such as signal loss, environmental interference, and equipment failure. For example, when a car travels through a tunnel, all satellite signals related to the car will be blocked by the tunnel, resulting in a long interval of consecutive missing values in the time series data.

Following the scheme in Khayati et al. (2020), we categorize the consecutive time series missing patterns into four distinct types: 'Disjoint', 'Overlap', 'MCAR_B', and 'Blackout'. Each pattern represents a specific form of long-interval consecutive missing data. This paper concentrates on the 'Blackout' pattern, considered the most challenging, where missing values across all channels are aligned in the same positions. This synchronized absence of data introduces significant difficulties for standard imputation methods, as it eliminates the possibility of leveraging information from other channels or time points to reconstruct the missing values. Figure 1 illustrates the differences between MCAR and Blackout missing patterns.

Traditional imputation methods perform poorly on this type of data due to their inadequate long-term modeling capabilities. Some methods attempt to address this problem in specific application domains (Ma et al., 2020; Chen et al., 2021; Guo et al., 2021), and other methods utilize mathematical and statistical techniques to tackle this issue (Wongoutong, 2020). However, these methods require domain knowledge and complex processing workflows, and they are not end-to-end deep learning solutions, thus limiting the range of problems they can effectively solve.

To address the problem of imputing Blackout data, we propose LSTI, a Long Short-Term Imputer specifically designed for this purpose. The model consists of three main components: first, a Long-Term Imputer, which is composed of a forward prediction network and a backward prediction network. During training, an additional consistency loss is introduced to minimize the discrepancy between the prediction of two networks. During imputation, the two prediction networks autoregressively impute the entire sequence in forward and backward directions, respectively, with the final imputation result obtained as a weighted sum of the two which follows the idea that the closer the values in the prediction window are to the lookback window, the smaller cumulative errors are. This structure effectively captures long-term dependencies in time series data and is suitable for imputing long-interval consecutive missing values.

The second component is the Short-Term Imputer. Missing data in real-world is often a mixture of various missing patterns, including MCAR and long-interval consecutive missing values. This requires a model that can adaptively impute different types of missing data. The Short-Term Imputer consists of a self-mapping network, trained with a randomly generated consecutive missing mask to capture short-term dependencies in the time series data. Subsequently, LSTI employs a Meta-weighting module to learn the importance of long-term and short-term dependencies in the current data, and uses this to weight the outputs of the long-term and short-term imputers to obtain the final imputation result. This structure can simultaneously learn both long-term and short-term dependencies in time series data, enabling adaptive and effective imputation across various missing data patterns.

Our main contributions are summarized as follows:

- We propose the Long-Term Imputer to capture long-term dependencies in time series data. It uses forward and backward prediction networks to autoregressively impute missing values from both directions. Additionally, a consistency loss is introduced during training to minimize the discrepancy between the two predictions.

- We propose the Short-Term Imputer to capture short-term dependencies in time series data. A Meta-weighting module is used to learn the proportion of long-term and short-term dependencies in specific data, adaptively balancing the weights of long-term and short-term dependencies. This allows LSTI to effectively impute data under various missing patterns.

- We conduct extensive experiments on five real-world datasets, and our method outperformed the current state-of-the-art imputation methods on all datasets.

## 2 Related Work

### 2.1 Time Series Imputation

Deep learning models are inherently good at capturing complex and non-linear relationships in data. RNN-based models are more commonly used in earlier works. GRU-D (Che et al., 2018), a variant of the Gated Recurrent Unit (GRU), addresses missing data in time series classification tasks. Soon after, BRITS (Cao et al., 2018) imputes missing values using a bidirectional recurrent dynamical system, without specific assumptions. In the case of M-RNN (Yoon et al., 2018), it imputes missing values based on hidden states derived from bidirectional RNNs. Since the generation ability is naturally suited to the imputation task, generative models are then widely used in filling missing values. E2GAN (Luo et al., 2019) incorporates a generator based on GRUI within an auto-encoder framework. For spatiotemporal sequence imputation, NAOMI (Liu et al., 2019) introduces a non-autoregressive model to comprise a bidirectional encoder and a multiresolution decoder. With diffusion models gaining popularity, CSDI (Tashiro et al., 2021) emerges, which is a conditional score-based diffusion model designed for time-series imputation. Advancing further, TimesNet (Wu et al., 2022) expands the examination of temporal variations into the two-dimensional space, while considering the presence of multiple periodicity in time series data. NRTSI (Shan et al., 2023) is an approach for time-series imputation that treats time series as a collection of (time, data) pairs. SAITS (Du et al., 2023) conducts simultaneous reconstruction and imputation by employing a weighted combination of two diagonally-masked self-attention blocks.

Each method discussed above possesses distinct advantages, but they are all designed for scenarios involving Missing Completely at Random (MCAR). When handling long-interval consecutive missing data, these methods may produce suboptimal results. Some methods attempt to address the imputation of consecutive missing data using specific domain knowledge in specific applications, such as air pollution (Ma et al., 2020), water quality prediction (Chen et al., 2021), and electrical engineering (Guo et al., 2021). Another method attempts to tackle this problem using a statistical approach (Wongoutong, 2020). However, these approaches require complex processing workflows and are not end-to-end methods, leading to limited applicability.

### 2.2 Time Series Forecasting

Time series forecasting methods potentially can be used to predict long-interval consecutive missing data. Specifically, the values prior to the missing interval are used to predict the entire missing segment. Some RNN-based models demonstrate robust forecasting performance in earlier studies, such as LSTNet (Lai et al., 2018), which combines networks of DNNs, RNNs, and Skip-RNNs. DeepAR (Salinas et al., 2020) utilizes an autoregressive recurrent network architecture to output the probabilities of prediction points. In more recent research, Transformer-based models gain significant prominence. Early on, Informer (Zhou et al., 2021) utilizes self-attention distilling to significantly accelerate the training efficiency of Transformers; following this, Autoformer (Wu et al., 2021) embeds time series decomposition into the model, enhancing its capability to discover temporal dependencies; Non-stationary Transformers (Liu et al., 2022) takes a novel approach by incorporating non-stationarity factors into the attention mechanism to prevent the over-stabilization of time series data; PatchTST (Nie et al., 2022) achieves remarkable results through the use of patches and

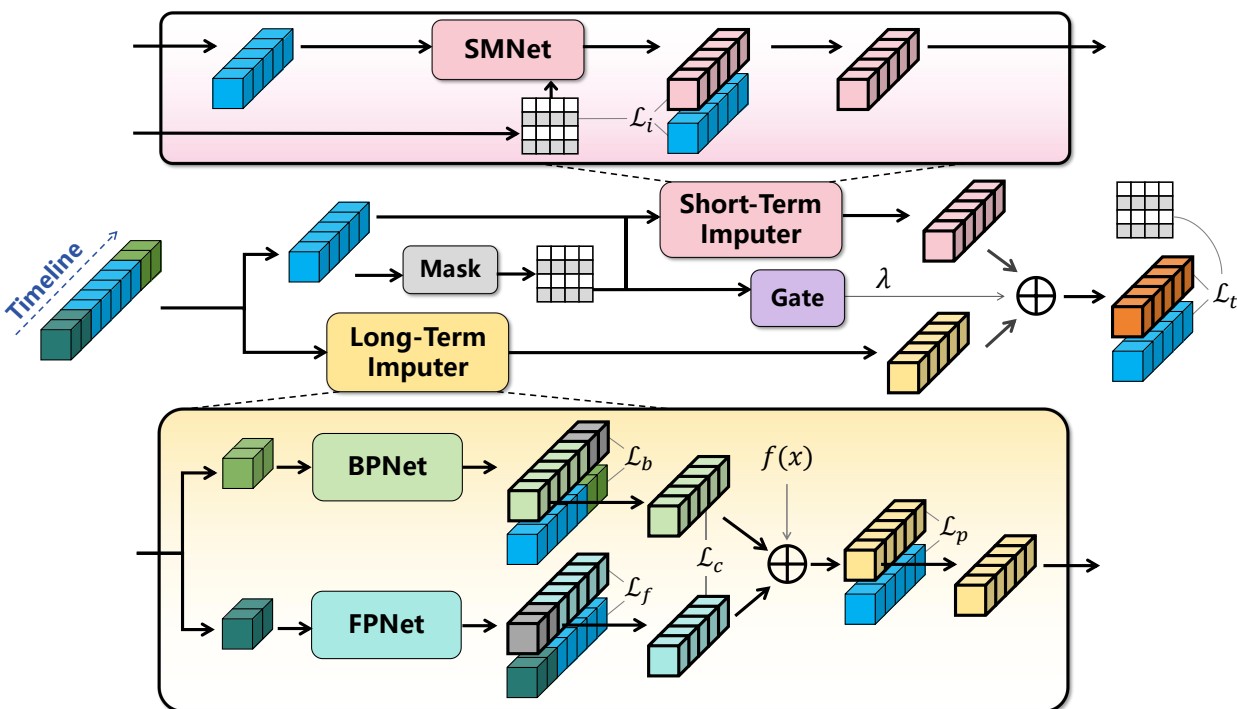

Figure 2: The overall training process of LSTI. LSTI consists of three main parts: (1) the Long-Term Imputer, which is composed of two prediction networks operating in different directions; (2) the Short-Term Imputer, which consists of a self-mapping module; (3) the Meta-weighting module, which learns to weigh the outputs of the two Imputers and can adapt to specific input data.

channel independence; most recently, iTransformer (Liu et al., 2023) transposes the input matrix and reverses the roles of the attention mechanism and feed-forward network, better capturing the correlations among multivariate time series.

Although forecasting methods appear promising for addressing long-interval consecutive missing data, various issues arise in practice. First, a series requiring imputation may have multiple missing points, and there is no guarantee that each missing segment is preceded by a complete, available segment for forecasting. Second, the endpoints of the predicted sequence cannot be effectively aligned with the observed values following the missing segment, leading to deviations in later forecasted values from actual values. Therefore, forecasting methods cannot be directly applied to impute long-interval consecutive missing data.

## 3 Long Short-Term Imputer

**Problem Formulation** We consider a collection of multivariate time series $\mathbf{X} = \{x_{1:T,1:C}\} \in \mathbb{R}^{T \times C}$ with $T$ timestamps and $C$ channels (attributes). The imputation task is to impute the missing values in $\mathbf{X}$. Formally, an observation mask is defined as $\mathbf{M} = \{m_{1:T,1:C}\} \in \mathbb{R}^{T \times C}$ where $m_{t,c} = 0$ if $x_{t,c}$ is missing, and $m_{t,c} = 1$ if $x_{t,c}$ is observed.

In practical scenarios, consecutive missing values can impact the matrix $\mathbf{M}$. The initial length of consecutive missing values is denoted as $L_m$. There are random submatrices $\mathbf{X}_{a_k:a_k+L_m-1,1:C} \subset \mathbf{X}$ where values are missing. Specifically, the observation mask can then be expressed as:

$$m_{t,c} = \prod_{k=1}^{K} \left(1 - \mathbb{1}(a_k \le t < a_k + L_m)\right), \quad m_{t,c} = \begin{cases} 0, & \text{if } x_{t,c} \text{ is missing,} \\ 1, & \text{if } x_{t,c} \text{ is observed.} \end{cases} \tag{1}$$

where $a_k$ represents the randomly selected starting position of the missing interval, and $K$ denotes the total number of missing intervals that appeared in the original data.

**Motivation**  Current imputation methods generally assume that test data is missing completely at random (MCAR). However, in the real world, missing values often occur consecutively and in clusters due to factors such as signal loss, environmental interference, and equipment failure. Mainstream imputation methods may not perform well when dealing with long-interval consecutive missing data because they are limited at learning long-term dependencies. Based on this practical concern, we design the Long Short-Term Imputer (LSTI), which can effectively impute consecutively missing data by learning both long-term dependencies and short-term dependencies with 2 specially designed expert models.

### 3.1  Overall structure

The LSTI model consists of the following modules: (1) Long-term Imputer: This module captures long-term dependencies in time series data. It includes forward and backward prediction networks, which use an additional consistency loss during training to minimize the discrepancy between them. During imputation, the forward and backward networks autoregressively impute the sequence in both directions, followed by a weighted ensemble for each missing interval. (2) Short-term Imputer: This module captures short-term dependencies in time series data. It includes a self-mapping imputation network, trained with a randomly generated consecutive missing mask. During imputation, it uses a sliding window approach for local imputation of the sequence. (3) Meta-weighting module: This module learns to weigh the outputs of the long-term and short-term Imputer considering the properties of specific input data. The final imputation result is obtained by ensembling the outputs of two imputers using the Meta-weighting approach. The overall framework of LSTI is shown in Figure 2. It is worth mentioning that the backbone of Long-term Imputer and Short-term Imputer can be any advanced backbone designed for time series, e.g., transformer-based model, RNN-based model or TCN-based model.

### 3.2  Long-term Imputer

The long-interval consecutive missing values may hinder the model from learning long dependencies. To alleviate the problem, two insights are employed to model long-term dependencies. First, the long-term dependencies can be learned in two directions. Thus Long-term Imputer includes a forward prediction network and a backward prediction network, which aim to learn long-term dependencies in each direction and impute missing data in an autoregressive manner. Second, the prediction results using the aforementioned two kinds of long-term dependencies should be consistent. As a result, during training, we incorporate a consistency loss to reduce the output discrepancy between the two networks to further enhance the effectiveness of learning long-term dependencies.

#### 3.2.1  Training process

**Bidirectional Prediction.**  Denote the input length of the prediction network as $S$ and the prediction length as $L$, for a training dataset $\mathbf{X}_{tr} \in \mathbb{R}^{(S+L+S)\times C}$ composed of input for forward prediction $\mathbf{X}_f \in \mathbb{R}^{S\times C}$, the ground-truth values to be predicted $\mathbf{X}_t \in \mathbb{R}^{L\times C}$, input for backward prediction $\mathbf{X}_b \in \mathbb{R}^{S\times C}$, we reverse time order of $\mathbf{X}_b$ to obtain $\mathbf{X}_b'$. Specifically, we have:

$$\mathbf{X}_{tr} = [\mathbf{X}_f, \mathbf{X}_t, \mathbf{X}_b], \quad (x_b')_{i,j} = (x_b)_{S-i,j}. \tag{2}$$

This indicates that $\mathbf{X}_f$ is a submatrix consisting of the first $S$ rows of $\mathbf{X}_{tr}$, while $\mathbf{X}_b'$ is a submatrix formed by reversing the last $S$ rows of $\mathbf{X}_{tr}$. $\mathbf{X}_f$ and $\mathbf{X}_b'$ are fed into the forward prediction network (FPNet) and the backward prediction network (BPNet), respectively, to obtain the network outputs $\mathbf{Y}_f, \mathbf{Y}_b' \in \mathbb{R}^{(S+L)\times C}$. $\mathbf{Y}_f$ consists of a predicted lookback window $\mathbf{Y}_{fl} \in \mathbb{R}^{S\times C}$ and a prediction window $\mathbf{Y}_{fp} \in \mathbb{R}^{L\times C}$, and similarly for $\mathbf{Y}_b'$. Specifically, we have:

$$\begin{aligned} \mathbf{Y}_f &= \text{FPNet}(\mathbf{X}_f), \quad \mathbf{Y}_b' = \text{BPNet}(\mathbf{X}_b'), \\ \mathbf{Y}_f &= [\mathbf{Y}_{fl}, \mathbf{Y}_{fp}], \qquad \mathbf{Y}_b' = [\mathbf{Y}_{bp}', \mathbf{Y}_{bl}'], \end{aligned} \tag{3}$$

where $\mathbf{Y}'_b$, $\mathbf{Y}'_{bl}$, $\mathbf{Y}'_{bp}$ is the reversal of $\mathbf{Y}_b$, $\mathbf{Y}_{bl}$, $\mathbf{Y}_{bp}$. The network outputs both the lookback window and the prediction window values to ensure that the model's output matches its input, resulting in a comprehensive training process and a more stable autoregressive process. Therefore, the forward prediction loss $\mathcal{L}_f$ and backward prediction loss $\mathcal{L}_b$ need to match both the lookback window and the prediction window values:

$$\mathcal{L}_f = \text{MSE}([\mathbf{X}_f, \mathbf{X}_t], \mathbf{Y}_f), \quad \mathcal{L}_b = \text{MSE}([\mathbf{X}_t, \mathbf{X}_b], \mathbf{Y}_b). \tag{4}$$

**Consistency Loss.** The prediction windows output $\mathbf{Y}_{fp}, \mathbf{Y}_{bp}$ by the two networks corresponding to the same segment, $\mathbf{X}_t$. If the long-term dependencies can be captured by two networks, their predicted values for the same segment should be as close as possible. As a result, we propose consistency loss to further reduce potential error accumulation in the autoregressive process:

$$\mathcal{L}_c = \text{MSE}(\mathbf{Y}_{fp}, \mathbf{Y}_{bp}). \tag{5}$$

**Emsembling.** Given predictions from both directions, we can ensemble them as the output of Long-term Imputer. An intuitive idea is that the closer the values in the prediction window are to the lookback window, the smaller cumulative errors are. Therefore, values at the beginning of the interval should be more aligned with the forward network's output, while values at the end of the interval should be more aligned with the backward network's output. We use a linear function $f(i)$ to weigh and ensemble the two prediction outputs, and compute the loss between the final sequence and the ground truth:

$$f(i) = \frac{i-1}{L-1}, \quad \mathbf{Y}_t = (1-f) \cdot \mathbf{Y}_{lp} + f \cdot \mathbf{Y}_{bp}, \quad \mathcal{L}_p = \text{MSE}(\mathbf{X}_t, \mathbf{Y}_t). \tag{6}$$

The overall training objective of Long-term Imputer $\mathcal{L}_l$ is the sum of the aforementioned losses:

$$\mathcal{L}_l = \mathcal{L}_f + \mathcal{L}_b + \mathcal{L}_c + \mathcal{L}_p. \tag{7}$$

### 3.2.2 Imputation process

In real-world datasets, there may be multiple consecutive missing segments, which can result in missing values in the network's input. To address this issue, the Long-term Imputer uses a bidirectional autoregressive approach to impute the entire missing data at once during the imputation process. This ensures that the data input to the network is either a true value or a previously imputed value, thereby avoiding the issue of missing data in the input.

**Bidirectional Autoregression.** Consider the case of the forward prediction network. We scan the entire dataset from front to back along the time dimension. If missing values are found in the data segment $\mathbf{D}_k \in \mathbb{R}^{L \times C}$, and the corresponding missing mask is $\mathbf{M}_k$, we select the preceding data segment $\mathbf{D}_{kl} \in \mathbb{R}^{S \times C}$ from $\mathbf{D}_k$ and input it into the forward prediction network:

$$[\mathbf{E}_{kl}, \mathbf{E}_{kp}] = \text{FPNet}(\mathbf{D}_{kl}), \quad \hat{\mathbf{D}}_k = \mathbf{M}_k \cdot \mathbf{D}_k + (1 - \mathbf{M}_k) \cdot \mathbf{E}_{kp}, \tag{8}$$

where $\hat{\mathbf{D}}_k$ is the imputed $\mathbf{D}_k$ sequence. We use $\hat{\mathbf{D}}_k$ to replace $\mathbf{D}_k$, ensuring that this sequence no longer contains any missing values. Then, we continue scanning the entire dataset and process the next segment $\mathbf{D}_{k+1}$.

The backward prediction network performs the same autoregressive imputation for the entire missing data from the opposite direction. Subsequently, we rescan the entire original missing data, and for each consecutive missing interval $\mathbf{H}_k \in \mathbb{R}^{L_k \times C}$, we emsemble the forward and backward imputation results using the method in Equation 6 to obtain the final imputed sequence $\hat{\mathbf{H}}$.

### 3.3 Short-term Imputer

Missing data in the real world is typically represented as a combination of single-point missing values and long consecutive missing intervals. To accommodate various missing patterns, we follow the idea of vanilla

imputation models and introduce a Short-term Imputer to capture short-term dependencies in time series data. This module consists of a single self-mapping network (SMNet). For the training data in Short-term Imputer, $\mathbf{X}_t \in \mathbb{R}^{L \times C}$, which is the middle segments of training data in Long-term Imputer (which is also illustrated in Figure 2), is used as input, and we randomly generate a consecutive missing mask $\mathbf{M}_t$ and input them into the self-mapping network:

$$\mathbf{Z}_t = \text{SMNet}(\mathbf{X}_t, \mathbf{M}_t), \quad \mathcal{L}_s = \text{MSE}((1 - \mathbf{M}_t) \cdot \mathbf{X}_t, (1 - \mathbf{M}_t) \cdot \mathbf{Z}_t). \tag{9}$$

During imputation, we partition the entire dataset using a sliding window of length $L$ and input the segments into the Short-term Imputer for imputation. Finally, the Short-term Imputer outputs the imputed data $\hat{\mathbf{I}}$.

### 3.4 Meta-weighting module

To adaptively impute data with a mixture of long-term and short-term missing values, we designed the Meta-weighting module. This module contains a standalone MLP that learns the characteristics of the missing data to adaptively allocate weights to the long-term and short-term imputer outputs. During training, after obtaining the output $\mathbf{Y}_t$ from the Long-term Imputer and $\mathbf{Z}_t$ from the Short-term Imputer, we input the data $\mathbf{X}_t$ and the missing mask $\mathbf{M}_t$ into the Meta-weighting module:

$$\lambda = \text{MLP}(\mathbf{X}_t, \mathbf{M}_t), \quad \hat{\mathbf{X}}_t = (1 - \lambda) \cdot \mathbf{Y}_t + \lambda \cdot \mathbf{Z}_t. \tag{10}$$

The training loss for Meta-weighting output is:

$$\mathcal{L}_t = \text{MSE}((1 - \mathbf{M}_t) \cdot \mathbf{X}_t, (1 - \mathbf{M}_t) \cdot \hat{\mathbf{X}}_t). \tag{11}$$

The overall training objective of LSTI is the sum of the losses from all modules:

$$\mathcal{L} = \mathcal{L}_l + \mathcal{L}_s + \mathcal{L}_t. \tag{12}$$

During imputation, after obtaining the output $\hat{\mathbf{H}}$ from the Long-term Imputer and $\hat{\mathbf{I}}$ from the Short-term Imputer, we use the same sliding window as the Short-term Imputer to get $\mathbf{D}_k$ and its missing mask $\mathbf{M}_k$ from the original data. These partitions are then input into the Meta-weighting module to obtain $\lambda_k$. For each window, we use the method in Equation 10 to meta-weighting $\hat{\mathbf{H}}_k$ and $\hat{\mathbf{I}}_k$ in proportion to get the final imputed data.

## 4 Experiments

In this section, we conduct experiments on five real-world datasets and compare them with the current mainstream imputation and forecasting methods. The experimental results indicate that our approach significantly outperforms existing methods in cases of long-interval consecutive missing data.

### 4.1 Experimental Setup

**Datasets**  We use five real-world datasets in experiments. (1) **Electricity** (UCI) collects hourly electricity consumption data of 321 customers from 2012 to 2014. (2) **Traffic** (PeMS) contains hourly road occupancy rates measured by 862 sensors on San Francisco Bay area freeways from January 2015 to December 2016. (3) **METR-LA** (Metro) records four months of statistics on traffic speed on 207 sensors on the highways of Los Angeles County. (4) **Guangzhou** (Chen et al., 2018) records traffic speeds per ten minutes on 214 anonymous roads in Guangzhou from August 1, 2016 to September 30, 2016. (5) **PEMS04** (Chen et al., 2001) is a subset of PEMS, collected by 307 detectors over a continuous period of 59 days, starting from January 1, 2018.

**Baselines**  We compare our method with several current mainstream imputation approaches, including **Transformer** (Vaswani et al., 2017), **SAITS** (Du et al., 2023), **TimesNet** (Wu et al., 2022), **CSDI** (Tashiro et al., 2021), and **ImputeFormer** (Nie et al., 2024). Additionally, we also conduct comparisons with long-term forecasting methods using **TimesNet** (Wu et al., 2022) and **iTransformer** (Liu et al., 2023) to validate the effectiveness of our approach.

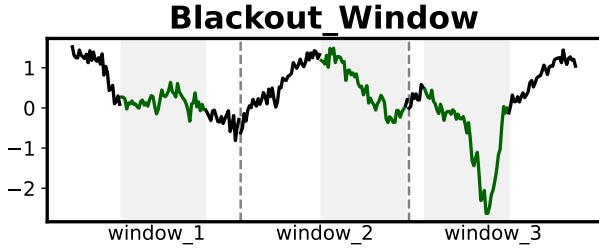
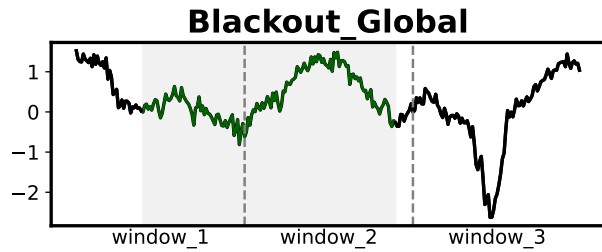

(a) Each window contains a gap of uniform length.     (b) The gaps can vary across windows.

Figure 3: Blackout missing patterns under $L_m = 50$ and $R_m = 50\%$. On the left, each window has a fixed missing segment of length 50. On the right, the missing segments are independent of the windows, potentially occurring at any position. The missing length and ratio may vary across windows.

**Generating missing data**   The Blackout missing pattern has been explored in previous studies (Khayati et al., 2020; Alcaraz & Strodthoff, 2022), but it is often constructed with missing data strictly aligned to imputation windows, a scenario that is uncommon in real-world datasets. In contrast, we propose a more realistic approach, where Blackout segments are randomly scattered across the entire dataset, independent of any specific imputation windows. As shown in Figure 3, we generate missing datasets using the following method: For a given missing length $L_m$ and a missing rate $R_m$, we randomly select intervals of length $L_m$ in the data and mark them as missing until the overall missing rate $R_m$ is reached. In our experiments, we conduct controlled variable comparisons by keeping either $L_m$ or $R_m$ constant while varying the other.

**Implementation details**   Our LSTI is a general training framework. Hence the backbone of FPNet, BPNet and SMNet can be any advanced deep network specially designed for time series. In our experiments, we use TimesNet as the backbone in LSTI, employing the same hyperparameter settings as in the original paper, namely $S$ is 96 and $L$ is 96. In other baseline models, hyperparameters are set according to the configurations specified in their original papers. We use MAE and MSE as the metrics for our experiments.

### 4.2   Main Results

To investigate the imputation capabilities of LSTI in various scenarios and its performance under different missing lengths $L_m$ and missing rates $R_m$, we conduct controlled variable comparisons by keeping either $L_m$ or $R_m$ constant while varying the other. Experimental results show that our method outperforms the current state-of-the-art models in all cases.

#### 4.2.1   Varying missing length

We investigate the imputation performance of different methods on real datasets with varying missing lengths, and the corresponding results are presented in Table 1. From the table, it can be observed that our method outperforms traditional imputation and forecasting methods across all datasets. For instance, regarding the MSE metric, our method leads to an average improvement of 72.32%, 55.43%, 40.27%, 54.54%, 68.29% compare to other methods on Electricity, Traffic, METR-LA, Guangzhou, and PEMS04 datasets, respectively. Additionally, we observe that as the length of consecutive missing data increases, the performance of traditional imputation and forecasting methods declines significantly, whereas our method exhibits a smaller decline across most datasets. This indicates that the length of consecutive missing data has a substantial impact on traditional imputation methods, whereas our method effectively imputes consecutive missing data and demonstrates better adaptability to datasets with consecutive missing values.

We conduct experiments on the Electricity dataset with varying lengths of consecutive missing data, and the results are shown in Figure 4a. As the length of consecutive missing data increases, the performance of all imputation methods declines to varying degrees. Models that perform reasonably well under low levels of consecutive missing data exhibit poor performance at higher levels. In contrast, our method is less impacted

Table 1: Imputation results on real datasets with different missing lengths and a fixed 30% missing rate. The label "_F" indicates models using forecasting methods.

| | $L_m$ | LSTI MAE | LSTI MSE | Transformer MAE | Transformer MSE | CSDI MAE | CSDI MSE | SAITS MAE | SAITS MSE | TimesNet MAE | TimesNet MSE | Imputeformer MAE | Imputeformer MSE | TimesNet_F MAE | TimesNet_F MSE | iTransformer_F MAE | iTransformer_F MSE |
|---|---|---|---|---|---|---|---|---|---|---|---|---|---|---|---|---|---|
| Electricity | 10 | **0.249** | **0.138** | 0.739 | 0.842 | 0.441 | 0.424 | 0.785 | 0.936 | 0.494 | 0.474 | 0.759 | 0.868 | 0.348 | 0.245 | 0.382 | 0.297 |
| | 30 | **0.242** | **0.129** | 0.782 | 0.923 | 0.546 | 0.616 | 0.798 | 0.965 | 0.571 | 0.618 | 0.788 | 0.926 | 0.330 | 0.222 | 0.372 | 0.290 |
| | 50 | **0.255** | **0.144** | 0.806 | 0.961 | 0.658 | 0.765 | 0.809 | 0.982 | 0.686 | 0.844 | 0.793 | 0.920 | 0.375 | 0.296 | 0.395 | 0.339 |
| | 100 | **0.278** | **0.171** | 0.830 | 1.029 | 0.774 | 1.005 | 0.828 | 1.041 | 0.784 | 1.008 | 0.814 | 0.976 | 0.433 | 0.407 | 0.434 | 0.414 |
| | 300 | **0.370** | **0.251** | 0.858 | 1.088 | 0.837 | 1.107 | 0.835 | 1.063 | 0.820 | 1.079 | 0.815 | 0.976 | 0.699 | 0.805 | 0.695 | 0.802 |
| Traffic | 10 | **0.269** | **0.296** | 0.743 | 1.129 | 0.590 | 0.816 | 0.762 | 1.158 | 0.603 | 0.830 | 0.739 | 1.143 | 0.374 | 0.456 | 0.401 | 0.497 |
| | 30 | **0.281** | **0.316** | 0.750 | 1.134 | 0.599 | 0.863 | 0.762 | 1.150 | 0.654 | 0.951 | 0.742 | 1.132 | 0.402 | 0.511 | 0.408 | 0.504 |
| | 50 | **0.301** | **0.341** | 0.762 | 1.164 | 0.555 | 0.871 | 0.769 | 1.166 | 0.697 | 1.056 | 0.750 | 1.124 | 0.389 | 0.478 | 0.373 | 0.454 |
| | 100 | **0.334** | **0.381** | 0.773 | 1.194 | 0.546 | 0.808 | 0.777 | 1.195 | 0.790 | 1.298 | 0.771 | 1.193 | 0.470 | 0.614 | 0.476 | 0.630 |
| | 300 | **0.450** | **0.547** | 0.782 | 1.213 | 0.635 | 0.972 | 0.782 | 1.210 | 0.785 | 1.240 | 0.769 | 1.207 | 0.663 | 0.986 | 0.670 | 1.002 |
| METR-LA | 10 | **0.342** | **0.353** | 0.674 | 1.119 | 0.411 | 0.826 | 0.580 | 0.881 | 0.460 | 0.636 | 0.677 | 1.006 | 0.565 | 0.848 | 0.644 | 1.074 |
| | 30 | **0.434** | **0.568** | 0.756 | 1.203 | 0.543 | 1.129 | 0.730 | 1.340 | 0.559 | 0.862 | 0.804 | 1.368 | 0.586 | 0.933 | 0.625 | 1.060 |
| | 50 | **0.508** | **0.747** | 0.795 | 1.272 | 0.586 | 1.345 | 0.742 | 1.381 | 0.559 | 0.926 | 0.773 | 1.216 | 0.592 | 0.951 | 0.612 | 0.987 |
| | 100 | **0.478** | **0.714** | 0.652 | 0.831 | 0.625 | 1.527 | 0.710 | 1.363 | 0.617 | 1.083 | 0.720 | 1.223 | 0.611 | 1.008 | 0.610 | 0.966 |
| | 300 | **0.712** | **1.225** | 0.842 | 1.567 | 0.768 | 1.729 | 0.811 | 1.658 | 0.812 | 1.476 | 0.773 | 1.365 | 0.817 | 1.519 | 0.828 | 1.527 |
| Guangzhou | 10 | **0.367** | **0.293** | 0.689 | 0.823 | 0.487 | 0.469 | 0.758 | 1.001 | 0.659 | 0.773 | 0.709 | 0.832 | 0.447 | 0.396 | 0.508 | 0.513 |
| | 30 | **0.387** | **0.323** | 0.698 | 0.845 | 0.606 | 0.678 | 0.747 | 0.975 | 0.797 | 1.080 | 0.714 | 0.871 | 0.449 | 0.411 | 0.512 | 0.528 |
| | 50 | **0.395** | **0.321** | 0.750 | 0.920 | 0.581 | 0.664 | 0.752 | 0.971 | 0.893 | 1.293 | 0.787 | 0.930 | 0.527 | 0.533 | 0.588 | 0.633 |
| | 100 | **0.420** | **0.350** | 0.748 | 0.909 | 0.588 | 0.672 | 0.752 | 0.938 | 0.828 | 1.126 | 0.762 | 0.915 | 0.540 | 0.545 | 0.559 | 0.584 |
| | 300 | **0.495** | **0.466** | 0.758 | 0.968 | 0.635 | 0.749 | 0.794 | 1.084 | 0.943 | 1.411 | 0.758 | 0.986 | 0.670 | 0.790 | 0.669 | 0.794 |
| PEMS04 | 10 | **0.213** | **0.099** | 0.662 | 0.622 | 0.294 | 0.160 | 0.714 | 0.741 | 0.461 | 0.389 | 0.820 | 0.908 | 0.485 | 0.495 | 0.649 | 0.808 |
| | 30 | **0.263** | **0.132** | 0.779 | 0.833 | 0.457 | 0.356 | 0.760 | 0.820 | 0.685 | 0.828 | 0.813 | 0.908 | 0.523 | 0.668 | 0.648 | 0.901 |
| | 50 | **0.267** | **0.142** | 0.827 | 0.897 | 0.553 | 0.505 | 0.762 | 0.813 | 0.711 | 0.865 | 0.854 | 0.965 | 0.531 | 0.611 | 0.531 | 0.532 |
| | 100 | **0.425** | **0.328** | 0.888 | 1.037 | 0.706 | 0.767 | 0.867 | 1.020 | 0.872 | 1.126 | 0.876 | 1.015 | 0.612 | 0.680 | 0.627 | 0.658 |
| | 300 | **0.569** | **0.513** | 0.883 | 1.025 | 0.731 | 0.785 | 0.899 | 1.097 | 0.891 | 1.167 | 0.923 | 1.096 | 0.776 | 0.876 | 0.800 | 0.906 |

by the length of consecutive missing data, exhibiting more stable performance across different lengths. It performs well under both low and high levels of consecutive missing data.

Furthermore, we observe that some models reach their worst performance even with very short lengths of consecutive missing data, whereas they perform much better under MCAR. This indicates that different models have varying levels of robustness to consecutive missing data. Some models experience a significant performance drop with only a small amount of consecutive missing data, which poses a substantial risk in real-world scenarios. However, our method demonstrates excellent robustness to consecutive missing data lengths and can effectively handle large-scale missing data problems in real-world applications.

### 4.2.2 Varying missing rate

In addition to handling data with different lengths of consecutive missing values, the imputation model should also adapt to data with varying missing rates. Table 2 presents the experimental results under different missing rates with a fixed missing length of 50. As shown, our method outperforms all baselines in every case. For instance, regarding the MSE metric, our method leads to an average improvement of 74.62%, 57.81%, 28.19%, 52.38%, 70.15% compare to other methods on Electricity, Traffic, METR-LA, Guangzhou, and PEMS04 datasets, respectively. This demonstrates that our model has strong robustness and can adapt to various types of missing data.

Additionally, we observe an anomalous phenomenon in some models where increasing the missing rate on certain datasets actually improves the results. This occurs because, at lower missing rates, the randomly consecutive missing values may be unevenly distributed, causing instability in the imputation models. Different random missing data can produce significantly varying results. If the missing intervals happen to concentrate

Table 2: Imputation results on real datasets with different missing rates and a fixed 50 missing length. The label "_F" indicates imputation using forecasting methods.

| | | LSTI | | Transformer | | CSDI | | SAITS | | TimesNet | | Imputeformer | | TimesNet_F | | iTransformer_F | |
|---|---|---|---|---|---|---|---|---|---|---|---|---|---|---|---|---|---|
| | $R_m$ | MAE | MSE | MAE | MSE | MAE | MSE | MAE | MSE | MAE | MSE | MAE | MSE | MAE | MSE | MAE | MSE |
| Electricity | 10% | **0.224** | **0.108** | 0.775 | 0.896 | 0.653 | 0.765 | 0.769 | 0.907 | 0.607 | 0.687 | 0.752 | 0.860 | 0.290 | 0.181 | 0.286 | 0.192 |
| | 30% | **0.255** | **0.144** | 0.806 | 0.961 | 0.658 | 0.766 | 0.809 | 0.982 | 0.686 | 0.844 | 0.793 | 0.920 | 0.375 | 0.296 | 0.395 | 0.339 |
| | 50% | **0.269** | **0.157** | 0.802 | 0.971 | 0.770 | 0.980 | 0.809 | 0.987 | 0.698 | 0.876 | 0.795 | 0.932 | 0.478 | 0.431 | 0.524 | 0.504 |
| | 70% | **0.321** | **0.209** | 0.863 | 1.121 | 0.766 | 0.979 | 0.821 | 1.019 | 0.733 | 0.945 | 0.803 | 0.940 | 0.578 | 0.572 | 0.637 | 0.668 |
| Traffic | 10% | **0.265** | **0.308** | 0.781 | 1.246 | 0.601 | 0.896 | 0.784 | 1.238 | 0.707 | 1.078 | 0.755 | 1.160 | 0.323 | 0.392 | 0.288 | 0.343 |
| | 30% | **0.301** | **0.341** | 0.762 | 1.164 | 0.555 | 0.871 | 0.769 | 1.166 | 0.697 | 1.056 | 0.756 | 1.170 | 0.389 | 0.478 | 0.373 | 0.454 |
| | 50% | **0.292** | **0.322** | 0.761 | 1.163 | 0.610 | 0.913 | 0.773 | 1.181 | 0.782 | 1.255 | 0.753 | 1.139 | 0.514 | 0.689 | 0.541 | 0.745 |
| | 70% | **0.366** | **0.428** | 0.766 | 1.166 | 0.619 | 0.961 | 0.767 | 1.156 | 0.826 | 1.397 | 0.773 | 1.244 | 0.604 | 0.833 | 0.629 | 0.877 |
| METR-LA | 10% | **0.478** | **0.710** | 0.770 | 1.099 | 0.509 | 0.938 | 0.704 | 1.266 | 0.594 | 0.988 | 0.690 | 1.003 | 0.638 | 1.249 | 0.577 | 0.928 |
| | 30% | **0.508** | **0.747** | 0.795 | 1.272 | 0.606 | 1.345 | 0.742 | 1.381 | 0.559 | 0.926 | 0.804 | 1.382 | 0.592 | 0.951 | 0.612 | 0.987 |
| | 50% | **0.568** | **0.867** | 0.744 | 1.210 | 0.626 | 1.375 | 0.727 | 1.351 | 0.589 | 1.048 | 0.788 | 1.344 | 0.674 | 1.150 | 0.672 | 1.093 |
| | 70% | **0.621** | **1.025** | 0.764 | 1.301 | 0.653 | 1.396 | 0.754 | 1.450 | 0.629 | 1.156 | 0.799 | 1.386 | 0.716 | 1.219 | 0.725 | 1.188 |
| Guangzhou | 10% | **0.410** | **0.355** | 0.766 | 1.024 | 0.557 | 0.607 | 0.800 | 1.107 | 0.856 | 1.246 | 0.712 | 0.867 | 0.443 | 0.399 | 0.481 | 0.491 |
| | 30% | **0.395** | **0.321** | 0.750 | 0.920 | 0.581 | 0.664 | 0.752 | 0.971 | 0.893 | 1.293 | 0.787 | 0.971 | 0.527 | 0.533 | 0.588 | 0.633 |
| | 50% | **0.428** | **0.368** | 0.733 | 0.904 | 0.563 | 0.631 | 0.766 | 1.007 | 0.888 | 1.306 | 0.757 | 0.950 | 0.595 | 0.639 | 0.637 | 0.728 |
| | 70% | **0.491** | **0.457** | 0.747 | 0.928 | 0.631 | 0.736 | 0.775 | 0.990 | 0.860 | 1.238 | 0.779 | 0.995 | 0.692 | 0.813 | 0.713 | 0.847 |
| PEMS04 | 10% | **0.232** | **0.109** | 0.819 | 0.915 | 0.447 | 0.319 | 0.780 | 0.834 | 0.728 | 0.861 | 0.854 | 0.965 | 0.281 | 0.160 | 0.366 | 0.279 |
| | 30% | **0.267** | **0.142** | 0.827 | 0.897 | 0.553 | 0.505 | 0.762 | 0.813 | 0.711 | 0.865 | 0.845 | 0.969 | 0.531 | 0.611 | 0.531 | 0.532 |
| | 50% | **0.349** | **0.235** | 0.812 | 0.901 | 0.622 | 0.628 | 0.799 | 0.895 | 0.719 | 0.921 | 0.863 | 0.994 | 0.743 | 1.040 | 0.737 | 0.940 |
| | 70% | **0.478** | **0.388** | 0.821 | 0.926 | 0.639 | 0.630 | 0.870 | 1.045 | 0.776 | 0.998 | 0.956 | 1.133 | 0.809 | 1.056 | 0.893 | 1.264 |

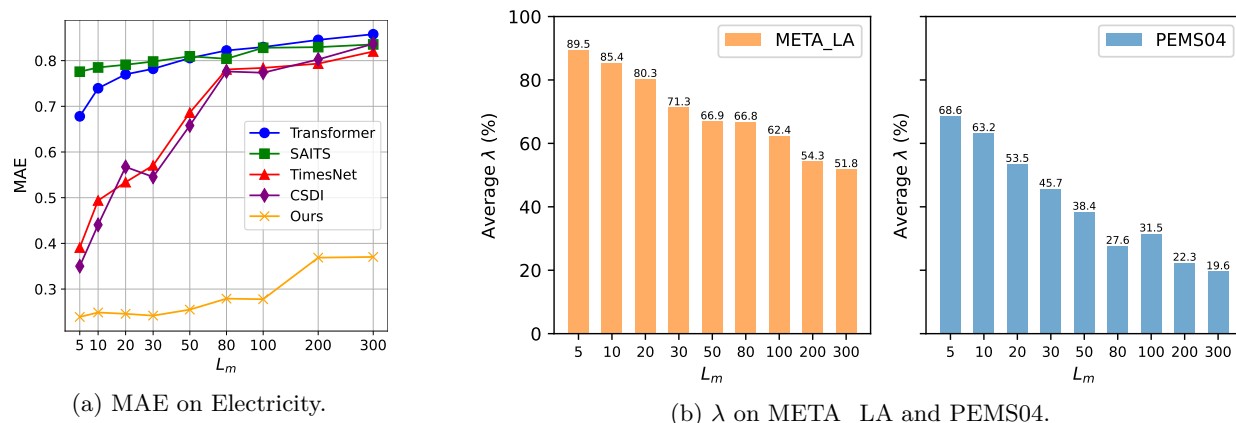

(a) MAE on Electricity.

(b) $\lambda$ on META_LA and PEMS04.

Figure 4: Results of MAE and average $\lambda$ with varying lengths of consecutive missing data.

within the model's imputation window, the model has to impute at a higher missing rate within that interval, leading to unexpectedly poor results.

### 4.3 Analysis $\lambda$

We test the output $\lambda$ of the Meta-weighting module on two datasets and record the average values of all $\lambda$ during testing. The relevant results are shown in Figure 4b. As can be seen, $\lambda$ generally decreases as $L_m$ increases, indicating that the Short-Term Imputer is more dominant in datasets with shorter $L_m$, whereas the Long-Term Imputer is more dominant in datasets with longer $L_m$. This is because shorter consecutive missing data can more easily capture information near the missing values, making short-term dependencies

more effective for imputation. Conversely, longer consecutive missing data can only obtain useful information from distant points, necessitating the use of long-term dependencies for better imputation. This demonstrates that our Meta-weighting module effectively learns the importance of long-term and short-term dependencies in different missing datasets and can adaptively adjust the weights based on the characteristics of the data.

### 4.4 Analysis data missing type

In real-world scenarios, various missing data patterns can occur. The straightforward Blackout missing pattern fails to fully capture real-world scenarios. In this section, we explore and conduct experiments under various missing patterns to assess the adaptability and effectiveness of LSTI across different settings.

#### 4.4.1 Analysis MCAR

Missing Completely At Random (MCAR) is a widely adopted missing data pattern in time-series imputation. It describes a scenario where the occurrence of missing data is independent of the data itself or any other variables. Under the MCAR pattern, missing data points are typically distributed uniformly across the dataset. Figure 5 illustrates the experimental results of various methods on the PEMS04 dataset under the MCAR missing pattern, with all models trained using the same MCAR configuration. As shown, despite not being specifically designed for MCAR, LSTI consistently achieves superior performance across different missing rates. This highlights the robustness and versatility of LSTI.

#### 4.4.2 Analysis other consecutive missing types

To evaluate the performance of LSTI under different consecutive missing patterns, we follow the scheme in Khayati et al. (2020), which categorizes these patterns into four types: 'Disjoint,' 'Overlap,' 'MCAR_B,' and 'Blackout.' Specifically, the 'Disjoint' pattern requires the missing blocks to have as little overlap as possible, while the 'Overlap' pattern demands the missing blocks to have as much overlap as possible. The 'MCAR_B' pattern involves randomly selecting missing blocks for each channel independently, whereas the 'Blackout' pattern randomly selects identical missing blocks across all channels. All experiments were conducted with $L_m$=50 and $R_m$=30% settings.

The experimental results are shown in Figure 6. It can be seen that our LSTI outperforms TimesNet across all datasets and missing patterns, demonstrating that our method has exceptional adaptability to various continuous missing patterns and exhibits high robustness, showing minimal sensitivity to the specific missing pattern. The figure also reveals that while TimesNet's performance remains consistent across the Disjoint, Overlap, and MCAR_B missing patterns, it significantly deteriorates under the Blackout missing pattern. This indicates that traditional imputation methods are less adaptable to the Blackout missing pattern, where missing values span all channels, making inference from other channels infeasible. In contrast, our LSTI

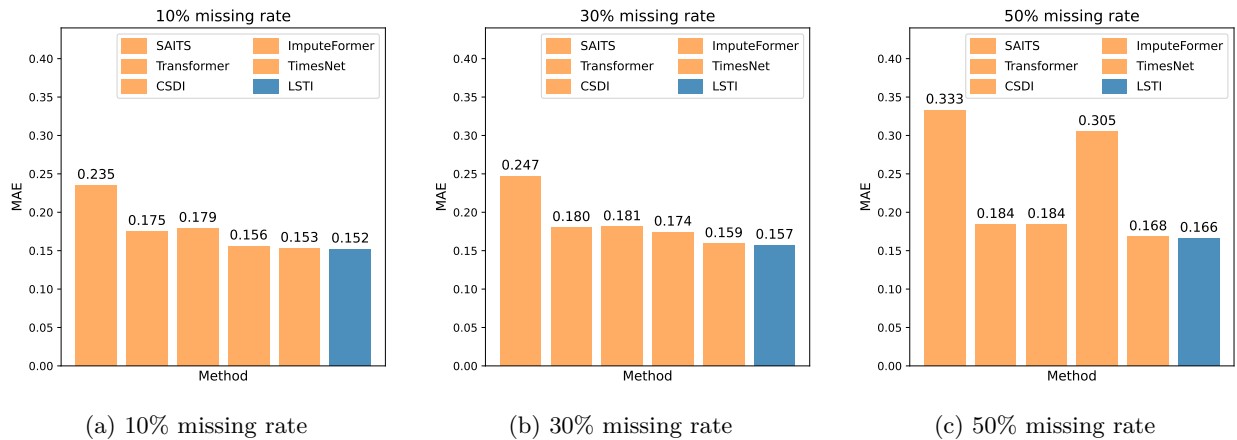

Figure 5: Experimental results of various methods on the PEMS04 dataset under the MCAR missing type.

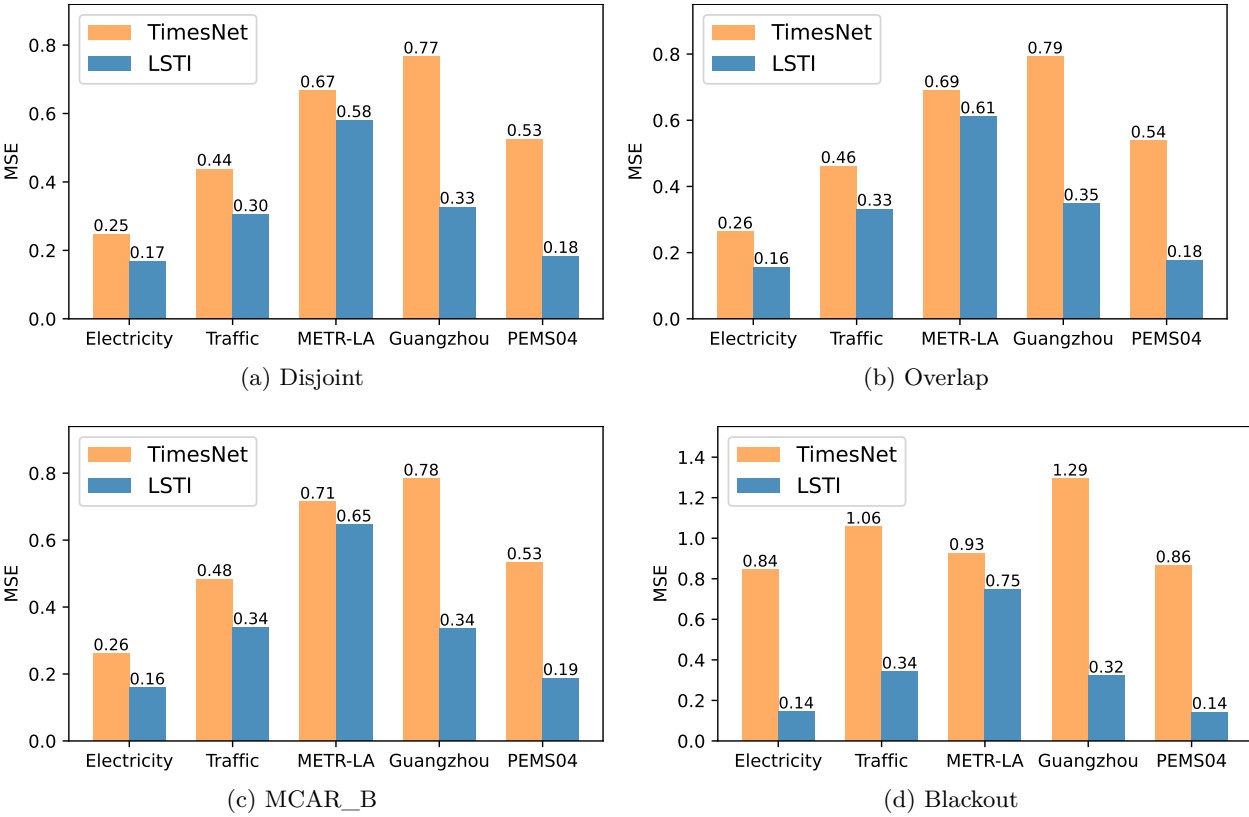

Figure 6: Comparison of MSE between TimesNet and LSTI across different datasets and missing patterns. All experiments were conducted with $L_m$=50 and $R_m$=30% settings.

Table 3: Ablation study results of different LSTI modules, with $L_m$ values of [10, 30, 50, 100, 300], and all results averaged over these five different $L_m$ values.

| fL | bL | L | S | LS | META-LA MAE | META-LA MSE | PEMS04 MAE | PEMS04 MSE |
|----|----|----|----|----|------|------|------|------|
| ✓ | - | - | - | - | 0.794 | 1.423 | 0.437 | 0.376 |
| - | ✓ | - | - | - | 0.692 | 1.362 | 0.413 | 0.367 |
| ✓ | ✓ | ✓ | - | - | 0.611 | 0.973 | 0.380 | 0.324 |
| - | - | - | ✓ | - | 0.558 | 0.963 | 0.476 | 0.435 |
| ✓ | ✓ | ✓ | ✓ | ✓ | **0.495** | **0.721** | **0.347** | **0.243** |

model effectively imputes data even under the challenging Blackout pattern, addressing a key limitation of traditional imputation approaches.

## 4.5 Ablation study

To investigate the impact of different LSTI modules on imputation performance, we conduct an ablation study on LSTI with the experimental results presented in Table 3. The abbreviations fL, bL, L, S, and LS represent the following configurations: only the forward prediction network, only the backward prediction network, only the Long-Term Imputer, only the Short-Term Imputer, and the complete LSTI model, respectively. We find that in all cases, the Long-Term Imputer outperformed both the forward and backward prediction networks

when used individually. This suggests that the strategy of weighted fusion of predictions made before and after the missing interval is both practical and highly effective.

Furthermore, while the Short-Term Imputer generally performed worse than the Long-Term Imputer on most datasets, their combined results achieved optimal performance. This is attributed to the fact that the Short-Term Imputer performs better when $L_m$ is small, but its performance significantly declines as $L_m$ increases. However, due to the presence of the Meta-weighting module, the negative impact of the Short-Term Imputer is minimized, resulting in the final averaged outcome being optimal. This demonstrates the importance of integrating both short-term and long-term components in the LSTI model for achieving robust imputation outcomes.

## 5 Conclusion

Long-interval consecutive missing data is a common issue in real-world time series datasets, which traditional deep learning imputation methods struggle to address effectively. To tackle this, we propose the Long Short-Term Imputer (LSTI), consisting of a Long-Term Imputer, a Short-Term Imputer, and a Meta-weighting module. This model can capture both long-term and short-term dependencies in time series and performs weighted merging based on data characteristics. Experiments on five real-world datasets demonstrate that our method can effectively impute long-interval consecutive missing data.

## Acknowledgements

This work was supported by the Shenzhen Science and Technology Program (ZDSYS20230626091203008), National Natural Science Foundation of China (62306085, 62302241, 62476071, 62236003), Shenzhen College Stability Support Plan (GXWD20231130151329002, GXWD20220817144428005).

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

# A   Variability and Confidence Estimation

## A.1   Standard deviation

The standard deviation ($s$) is a key statistical metric used to quantify the dispersion or spread of a data set. It measures the extent to which individual data points deviate from the mean, offering valuable insight into the variability or consistency of the data. A larger standard deviation indicates greater variability, with data points more widely dispersed from the mean, whereas a smaller standard deviation signifies less variability, suggesting that the data points are more tightly clustered around the mean.

$$\bar{x} = \frac{1}{n}\sum_{i=1}^{n} x_i, \quad s = \sqrt{\frac{1}{n-1}\sum_{i=1}^{n}(x_i - \bar{x})^2} \tag{13}$$

In our experiment, each result is the average of three repeated trials, i.e., $n = 3$. Table 4 and 5 present the standard deviations for all experimental results. It can be observed that, in the majority of experimental settings, LSTI exhibits a relatively small standard deviation, which further validates the consistency and stability of our findings.

Table 4: Standard deviation on real datasets with different missing lengths and a fixed 30% missing rate. The label "_F" indicates models using forecasting methods.

| | $L_m$ | LSTI MAE | LSTI MSE | CSDI MAE | CSDI MSE | TimesNet MAE | TimesNet MSE | SAITS MAE | SAITS MSE | Transformer MAE | Transformer MSE | TimesNet_F MAE | TimesNet_F MSE | iTransformer_F MAE | iTransformer_F MSE |
|---|---|---|---|---|---|---|---|---|---|---|---|---|---|---|---|
| Electricity | 10 | 0.008 | 0.011 | 0.094 | 0.145 | 0.031 | 0.059 | 0.005 | 0.007 | 0.018 | 0.038 | 0.017 | 0.032 | 0.007 | 0.018 |
| | 30 | 0.006 | 0.010 | 0.021 | 0.028 | 0.038 | 0.072 | 0.009 | 0.025 | 0.023 | 0.057 | 0.001 | 0.002 | 0.011 | 0.009 |
| | 50 | 0.003 | 0.007 | 0.023 | 0.033 | 0.032 | 0.082 | 0.011 | 0.025 | 0.014 | 0.035 | 0.002 | 0.010 | 0.023 | 0.026 |
| | 100 | 0.022 | 0.028 | 0.027 | 0.047 | 0.025 | 0.037 | 0.014 | 0.016 | 0.013 | 0.036 | 0.026 | 0.029 | 0.029 | 0.021 |
| | 300 | 0.025 | 0.025 | 0.014 | 0.045 | 0.027 | 0.137 | 0.002 | 0.019 | 0.012 | 0.032 | 0.033 | 0.056 | 0.033 | 0.056 |
| Traffic | 10 | 0.005 | 0.016 | 0.003 | 0.011 | 0.015 | 0.018 | 0.012 | 0.040 | 0.014 | 0.041 | 0.021 | 0.060 | 0.017 | 0.044 |
| | 30 | 0.008 | 0.019 | 0.025 | 0.052 | 0.030 | 0.059 | 0.014 | 0.036 | 0.013 | 0.043 | 0.020 | 0.071 | 0.018 | 0.046 |
| | 50 | 0.028 | 0.038 | 0.025 | 0.050 | 0.037 | 0.099 | 0.015 | 0.064 | 0.014 | 0.067 | 0.042 | 0.068 | 0.018 | 0.026 |
| | 100 | 0.023 | 0.031 | 0.008 | 0.046 | 0.015 | 0.073 | 0.011 | 0.051 | 0.012 | 0.047 | 0.048 | 0.081 | 0.049 | 0.086 |
| | 300 | 0.031 | 0.073 | 0.042 | 0.108 | 0.013 | 0.054 | 0.010 | 0.042 | 0.007 | 0.037 | 0.039 | 0.075 | 0.041 | 0.083 |
| METR-LA | 10 | 0.024 | 0.058 | 0.070 | 0.288 | 0.034 | 0.078 | 0.116 | 0.397 | 0.100 | 0.254 | 0.042 | 0.155 | 0.051 | 0.093 |
| | 30 | 0.018 | 0.061 | 0.076 | 0.274 | 0.026 | 0.100 | 0.046 | 0.179 | 0.044 | 0.158 | 0.057 | 0.231 | 0.032 | 0.043 |
| | 50 | 0.004 | 0.027 | 0.053 | 0.215 | 0.055 | 0.207 | 0.037 | 0.115 | 0.013 | 0.075 | 0.023 | 0.115 | 0.044 | 0.181 |
| | 100 | 0.115 | 0.352 | 0.122 | 0.450 | 0.050 | 0.222 | 0.092 | 0.261 | 0.087 | 0.297 | 0.027 | 0.100 | 0.046 | 0.217 |
| | 300 | 0.131 | 0.280 | 0.188 | 0.790 | 0.102 | 0.370 | 0.193 | 0.657 | 0.179 | 0.642 | 0.182 | 0.602 | 0.171 | 0.591 |
| Guangzhou | 10 | 0.006 | 0.011 | 0.065 | 0.078 | 0.030 | 0.062 | 0.018 | 0.052 | 0.034 | 0.061 | 0.023 | 0.032 | 0.014 | 0.017 |
| | 30 | 0.050 | 0.082 | 0.086 | 0.121 | 0.019 | 0.075 | 0.021 | 0.092 | 0.021 | 0.064 | 0.026 | 0.049 | 0.007 | 0.026 |
| | 50 | 0.007 | 0.028 | 0.031 | 0.074 | 0.082 | 0.168 | 0.025 | 0.073 | 0.021 | 0.041 | 0.023 | 0.038 | 0.037 | 0.029 |
| | 100 | 0.053 | 0.055 | 0.032 | 0.063 | 0.008 | 0.014 | 0.013 | 0.032 | 0.010 | 0.040 | 0.077 | 0.121 | 0.069 | 0.110 |
| | 300 | 0.009 | 0.023 | 0.036 | 0.026 | 0.106 | 0.294 | 0.033 | 0.084 | 0.019 | 0.047 | 0.014 | 0.012 | 0.016 | 0.009 |
| PEMS04 | 10 | 0.006 | 0.011 | 0.024 | 0.025 | 0.005 | 0.006 | 0.016 | 0.027 | 0.008 | 0.012 | 0.118 | 0.238 | 0.028 | 0.108 |
| | 30 | 0.023 | 0.015 | 0.052 | 0.082 | 0.029 | 0.053 | 0.051 | 0.085 | 0.073 | 0.117 | 0.056 | 0.223 | 0.032 | 0.205 |
| | 50 | 0.027 | 0.037 | 0.037 | 0.076 | 0.024 | 0.072 | 0.027 | 0.050 | 0.026 | 0.044 | 0.029 | 0.094 | 0.028 | 0.035 |
| | 100 | 0.061 | 0.098 | 0.049 | 0.108 | 0.035 | 0.054 | 0.047 | 0.084 | 0.040 | 0.070 | 0.124 | 0.236 | 0.118 | 0.181 |
| | 300 | 0.080 | 0.138 | 0.042 | 0.100 | 0.027 | 0.103 | 0.018 | 0.065 | 0.011 | 0.018 | 0.039 | 0.059 | 0.026 | 0.037 |

Table 5: Standard deviation on real datasets with different missing rates and a fixed 50 missing length. The label "_F" indicates imputation using forecasting methods.

| | $R_m$ | LSTI MAE | LSTI MSE | CSDI MAE | CSDI MSE | TimesNet MAE | TimesNet MSE | SAITS MAE | SAITS MSE | Transformer MAE | Transformer MSE | TimesNet_F MAE | TimesNet_F MSE | iTransformer_F MAE | iTransformer_F MSE |
|---|---|---|---|---|---|---|---|---|---|---|---|---|---|---|---|
| Electricity | 10% | 0.003 | 0.003 | 0.092 | 0.158 | 0.030 | 0.055 | 0.023 | 0.052 | 0.034 | 0.079 | 0.014 | 0.017 | 0.012 | 0.015 |
| | 30% | 0.003 | 0.007 | 0.038 | 0.095 | 0.032 | 0.082 | 0.011 | 0.025 | 0.014 | 0.035 | 0.002 | 0.010 | 0.023 | 0.026 |
| | 50% | 0.007 | 0.011 | 0.044 | 0.101 | 0.016 | 0.049 | 0.008 | 0.012 | 0.031 | 0.053 | 0.021 | 0.036 | 0.006 | 0.015 |
| | 70% | 0.018 | 0.023 | 0.035 | 0.070 | 0.018 | 0.033 | 0.007 | 0.016 | 0.023 | 0.046 | 0.020 | 0.038 | 0.015 | 0.027 |
| Traffic | 10% | 0.014 | 0.009 | 0.084 | 0.179 | 0.049 | 0.115 | 0.021 | 0.049 | 0.020 | 0.061 | 0.033 | 0.051 | 0.052 | 0.074 |
| | 30% | 0.028 | 0.038 | 0.025 | 0.050 | 0.037 | 0.099 | 0.015 | 0.064 | 0.014 | 0.067 | 0.042 | 0.068 | 0.018 | 0.026 |
| | 50% | 0.012 | 0.018 | 0.075 | 0.105 | 0.026 | 0.088 | 0.011 | 0.051 | 0.016 | 0.055 | 0.070 | 0.162 | 0.025 | 0.060 |
| | 70% | 0.029 | 0.046 | 0.065 | 0.109 | 0.018 | 0.029 | 0.007 | 0.014 | 0.014 | 0.021 | 0.023 | 0.046 | 0.027 | 0.050 |
| METR-LA | 10% | 0.164 | 0.393 | 0.071 | 0.246 | 0.171 | 0.534 | 0.191 | 0.653 | 0.171 | 0.525 | 0.238 | 0.767 | 0.144 | 0.469 |
| | 30% | 0.004 | 0.027 | 0.053 | 0.215 | 0.055 | 0.207 | 0.037 | 0.115 | 0.013 | 0.075 | 0.023 | 0.115 | 0.044 | 0.181 |
| | 50% | 0.050 | 0.164 | 0.046 | 0.181 | 0.034 | 0.105 | 0.037 | 0.174 | 0.014 | 0.101 | 0.029 | 0.070 | 0.048 | 0.168 |
| | 70% | 0.034 | 0.134 | 0.040 | 0.178 | 0.041 | 0.096 | 0.003 | 0.020 | 0.016 | 0.024 | 0.046 | 0.145 | 0.027 | 0.039 |
| Guangzhou | 10% | 0.020 | 0.048 | 0.090 | 0.148 | 0.039 | 0.089 | 0.048 | 0.133 | 0.048 | 0.110 | 0.013 | 0.041 | 0.054 | 0.134 |
| | 30% | 0.007 | 0.028 | 0.031 | 0.074 | 0.082 | 0.168 | 0.025 | 0.073 | 0.021 | 0.041 | 0.023 | 0.038 | 0.037 | 0.029 |
| | 50% | 0.019 | 0.026 | 0.040 | 0.075 | 0.024 | 0.043 | 0.022 | 0.053 | 0.007 | 0.024 | 0.029 | 0.048 | 0.015 | 0.041 |
| | 70% | 0.008 | 0.002 | 0.016 | 0.053 | 0.036 | 0.114 | 0.006 | 0.005 | 0.011 | 0.017 | 0.039 | 0.091 | 0.022 | 0.038 |
| PEMS04 | 10% | 0.056 | 0.042 | 0.046 | 0.053 | 0.043 | 0.094 | 0.032 | 0.049 | 0.046 | 0.093 | 0.037 | 0.024 | 0.027 | 0.040 |
| | 30% | 0.027 | 0.037 | 0.037 | 0.076 | 0.024 | 0.072 | 0.027 | 0.050 | 0.026 | 0.044 | 0.029 | 0.094 | 0.028 | 0.035 |
| | 50% | 0.046 | 0.072 | 0.065 | 0.103 | 0.047 | 0.136 | 0.032 | 0.056 | 0.033 | 0.049 | 0.055 | 0.280 | 0.036 | 0.121 |
| | 70% | 0.008 | 0.013 | 0.067 | 0.102 | 0.035 | 0.055 | 0.049 | 0.135 | 0.007 | 0.008 | 0.072 | 0.227 | 0.039 | 0.138 |

Table 6: The 95% confidence intervals of LSTI under a fixed 30% missing rate across all datasets. Here, [x,y] indicates the confidence interval, where x is the lower bound and y is the upper bound.

| $L_m$ | Electricity MAE | Electricity MSE | Traffic MAE | Traffic MSE | Metr-la MAE | Metr-la MSE | GuangZhou MAE | GuangZhou MSE | PEMS04 MAE | PEMS04 MSE |
|---|---|---|---|---|---|---|---|---|---|---|
| 10 | [0.24, 0.25] | [0.13, 0.14] | [0.29, 0.35] | [0.30, 0.43] | [0.40, 0.47] | [0.46, 0.65] | [0.39, 0.42] | [0.32, 0.37] | [0.23, 0.26] | [0.12, 0.15] |
| 30 | [0.24, 0.24] | [0.12, 0.13] | [0.28, 0.47] | [0.32, 0.58] | [0.45, 0.54] | [0.57, 0.77] | [0.35, 0.43] | [0.27, 0.38] | [0.23, 0.27] | [0.10, 0.19] |
| 50 | [0.25, 0.26] | [0.14, 0.16] | [0.30, 0.36] | [0.32, 0.44] | [0.47, 0.59] | [0.66, 0.93] | [0.38, 0.43] | [0.29, 0.39] | [0.23, 0.31] | [0.10, 0.22] |
| 100 | [0.25, 0.30] | [0.14, 0.20] | [0.34, 0.40] | [0.38, 0.47] | [0.40, 0.67] | [0.38, 1.27] | [0.39, 0.49] | [0.33, 0.43] | [0.28, 0.46] | [0.15, 0.39] |
| 300 | [0.37, 0.43] | [0.25, 0.32] | [0.45, 0.56] | [0.52, 0.77] | [0.63, 0.89] | [1.02, 1.53] | [0.50, 0.52] | [0.47, 0.49] | [0.47, 0.56] | [0.34, 0.50] |

## A.2 Confidence interval

A confidence interval (CI) is a statistical tool used to estimate the range within which a true population parameter is likely to lie, based on sample data. It quantifies the uncertainty of the estimate, providing both an estimate of the population mean and an indication of the precision of that estimate. A 95% confidence interval, for example, means that if the same experiment were repeated multiple times, approximately 95% of the calculated confidence intervals would contain the true population mean, offering a measure of the reliability of the estimate. The key benefits of calculating a CI include quantifying uncertainty, enabling more informed decision-making, and assessing the statistical significance of experimental results. The confidence interval for the sample mean is calculated using the formula:

$$\text{CI} = \bar{x} \pm z \cdot \frac{s}{\sqrt{n}} \tag{14}$$

Table 7: The 95% confidence intervals of LSTI under a fixed 50 missing length across all datasets. Here, [x,y] indicates the confidence interval, where x is the lower bound and y is the upper bound.

| $R_m$ | Electricity | | Traffic | | Metr-la | | GuangZhou | | PEMS04 | |
| --- | --- | --- | --- | --- | --- | --- | --- | --- | --- | --- |
| | MAE | MSE | MAE | MSE | MAE | MSE | MAE | MSE | MAE | MSE |
| 0.1 | [0.22, 0.23] | [0.10, 0.11] | [0.25, 0.28] | [0.30, 0.32] | [0.29, 0.66] | [0.27, 1.16] | [0.39, 0.43] | [0.30, 0.41] | [0.17, 0.30] | [0.06, 0.16] |
| 0.3 | [0.25, 0.26] | [0.14, 0.15] | [0.27, 0.33] | [0.30, 0.38] | [0.50, 0.51] | [0.72, 0.78] | [0.39, 0.40] | [0.29, 0.35] | [0.24, 0.30] | [0.10, 0.18] |
| 0.5 | [0.26, 0.28] | [0.14, 0.17] | [0.28, 0.31] | [0.30, 0.34] | [0.51, 0.63] | [0.68, 1.05] | [0.41, 0.45] | [0.34, 0.40] | [0.30, 0.40] | [0.15, 0.32] |
| 0.7 | [0.30, 0.34] | [0.18, 0.24] | [0.33, 0.40] | [0.38, 0.48] | [0.58, 0.66] | [0.87, 1.18] | [0.48, 0.50] | [0.45, 0.46] | [0.47, 0.49] | [0.37, 0.40] |

where $\bar{x}$ represents the sample mean, $s$ is the sample standard deviation, $n$ denotes the sample size ($n = 3$ in our experiment), and $z$ is the critical value corresponding to the desired confidence level (1.96 for $n = 3$ and a 95% CI). Table 6 and 7 illustrate the 95% confidence intervals for LSTI across all datasets, providing a comprehensive view of its performance variability.

### A.3 Cumulative Average

To examine how many repetitions are needed for the average result to converge, we conducted 30 repeated experiments on the GuangZhou dataset with $L_m = 50$ and $R_m = 30\%$. The results are shown in Figure 7, where the x-axis represents the number of repeated experiments and the y-axis represents the average MAE of the first x experiments. The average MAE across all 30 experiments is shown as a dashed line. As observed, the average value converges to the overall mean after approximately the 5th experiment, with subsequent fluctuations remaining small around the mean. This demonstrates that our method exhibits strong stability.

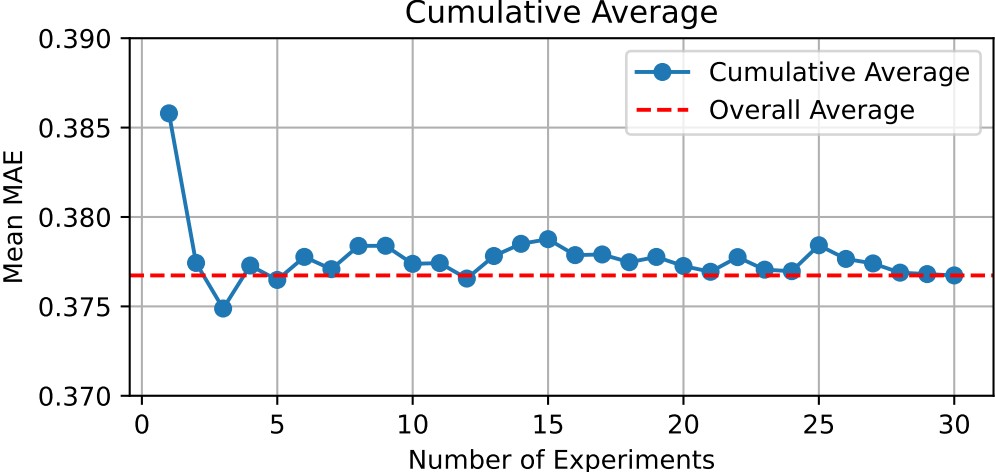

Figure 7: Repeated experiments conducted on the GuangZhou dataset with $L_m = 50$ and $R_m = 30\%$, where the missing data mask is identical for each experiment.

# B  Ablation Study on Loss Function

In the Long-term Imputer, we utilize four distinct loss functions: forward prediction loss $\mathcal{L}_f$, backward prediction loss $\mathcal{L}_b$, consistency loss $\mathcal{L}_c$, and combined result loss $\mathcal{L}_p$. They play an indispensable role in the training process. In the following, we provide a detailed description of each loss function, along with the corresponding ablation study results.

## B.1  Detailed explanation

The forward prediction loss $\mathcal{L}_f$ and backward prediction loss $\mathcal{L}_b$ serve as the direct loss functions for FPNet and BPNet, respectively. Their primary distinction lies in the prediction direction of the networks. For instance, $\mathcal{L}_f$ calculates the loss for the prediction window generated by FPNet while simultaneously computing the reconstruction loss for the look-back window produced by FPNet. This design improves the stability of the Long-term Imputer during autoregressive processes and reduces the risk of significant error accumulation over successive iterations. These two loss functions directly govern the training of FPNet and BPNet, aiming to optimize the values generated by both networks.

The consistency loss $\mathcal{L}_c$ is applied to both FPNet and BPNet. As shown in Figure 2, during training, the prediction windows generated by FPNet and BPNet correspond to the same segment of the time series. Ideally, their predictions should be identical. The consistency loss $\mathcal{L}_c$ measures the difference between the prediction windows produced by FPNet and BPNet, with the goal of aligning their prediction spaces and ensuring consistent outputs. This design allows the Long-term Imputer to produce stable results when imputing the same target segment, whether in the forward or backward direction, thereby mitigating the instability often associated with autoregressive processes.

The loss $\mathcal{L}_p$ is associated with the final combined result produced by the Long-term Imputer. This is the most crucial loss, as it directly reflects the imputation capability of the Long-term Imputer. The individual performance of FPNet and BPNet does not necessarily translate to optimal imputation performance, as the prediction windows they generate must be combined using a scalar weighting mechanism to produce the final imputed result. The role of $\mathcal{L}_p$ is to integrate this mechanism into the network, aiming to optimize the combined outputs of FPNet and BPNet and, in turn, directly enhance the imputation capability of the Long-term Imputer.

Table 8: Ablation study on different losses with a fixed 50 missing length and 30% missing rate.

| $\mathcal{L}_f$ | $\mathcal{L}_b$ | $\mathcal{L}_c$ | $\mathcal{L}_p$ | GuangZhou | | Electricity | |
|---|---|---|---|---|---|---|---|
| | | | | MAE | MSE | MAE | MSE |
| ✓ | ✓ | ✓ | ✓ | **0.395** | **0.323** | **0.272** | **0.154** |
| ✓ | ✓ | - | - | 0.426 | 0.357 | 0.340 | 0.278 |
| ✓ | ✓ | ✓ | - | 0.427 | 0.373 | 0.347 | 0.234 |
| ✓ | ✓ | - | ✓ | 0.418 | 0.355 | 0.292 | 0.181 |
| - | - | ✓ | ✓ | 0.423 | 0.375 | 0.293 | 0.190 |
| - | - | - | ✓ | 0.440 | 0.382 | 0.296 | 0.184 |

## B.2  Ablation study

We conducte ablation experiments on different losses across various datasets. In these experiments, we modify the training loss $\mathcal{L}_l$ of the Long-term Imputer by combining different losses to investigate their individual contributions. The training processes of other modules, such as the Short-term Imputer and the meta-learning weighting module, remain unchanged.

The results of the experiments are shown in Table 8. As observed, the combination of all four losses in $\mathcal{L}_l$ consistently produced the best performance across all datasets. This demonstrates that each of the four losses plays a crucial and complementary role in the training process.

# C   Details of the Short-term Imputer

The Short-term Imputer is a crucial component of LSTI, designed to capture short-term dependencies in time series data. It is particularly effective in datasets with densely distributed observation points. While the main text describes the Short-term Imputer as a self-mapping network (SMNet), this section offers a more detailed explanation, covering both its training and inference processes.

## C.1   The training process of Short-term Imputer

At each training step, a data segment $\mathbf{X}_{tr} \in \mathbb{R}^{(S+L+S)\times C}$ is obtained and divided into three parts: $\mathbf{X}_{tr} = [\mathbf{X}_f, \mathbf{X}_t, \mathbf{X}_b]$. The Short-term Imputer is trained exclusively on the middle segment, $\mathbf{X}_t \in \mathbb{R}^{L\times C}$. At the same time, we obtain the corresponding mask matrix $\mathbf{M}_t$ for $\mathbf{X}_t$. We first normalize the valid values in each channel of $\mathbf{X}_t$, and set all invalid values to 0. The processing formula is as follows:

$$\mu_c = \frac{\sum_{t=1}^{L} (m_{t,c} \cdot x_{t,c})}{\sum_{t=1}^{L} m_{t,c}}, \quad \sigma_c^2 = \frac{\sum_{t=1}^{L} (x_{t,c} - \mu_c)^2 \cdot m_{t,c}}{\sum_{i=1}^{L} m_{t,c}}, \quad \bar{x}_{t,c} = \frac{1}{\sigma_c} \cdot (x_{t,c} - \mu_c) \cdot m_{t,c}. \tag{15}$$

The normalization method described above excludes zero-filled missing values, using only valid observations to compute the mean and standard deviation. This reduces the negative impact of zero values, ensuring that the computed statistics are more representative of the ground truth distribution and improving the accuracy and reliability of the imputation process.

The normalized matrix $\bar{\mathbf{X}}_t$ is transformed into feature tokens via a linear fully connected layer and then summed with a fully trainable positional encoding matrix to preserve the temporal order of the time series. The detailed data encoding process is described as follows:

$$\mathbf{X}_{enc} = \bar{\mathbf{X}}_t \mathbf{W}_{enc} + \mathbf{B}_{enc} + \mathbf{E}_{pos}, \tag{16}$$

where $\mathbf{W}_{enc}$, $\mathbf{B}_{enc}$, and $\mathbf{E}_{pos}$ are learnable parameters. The projection sequence $\mathbf{X}_{enc}$ constitutes the input sequence of the SMNet. We then feed $\mathbf{X}_{enc}$ into the SMNet to obtain the imputed result $\bar{\mathbf{Z}}_t$ produced by the network. Subsequently, $\bar{\mathbf{Z}}_t$ is de-normalized using the previously obtained mean $\mu$ and standard deviation $\sigma$ to obtain the final result. The Short-term Imputer loss $\mathcal{L}_s$ is then computed based on this final output.

$$\bar{\mathbf{Z}}_t = \text{SMNet}(\mathbf{X}_{enc}), \quad \mathbf{Z}_t = \bar{\mathbf{Z}}_t \cdot \sigma + \mu, \quad \mathcal{L}_s = \text{MSE}((1 - \mathbf{M}_t) \cdot \mathbf{X}_t, (1 - \mathbf{M}_t) \cdot \mathbf{Z}_t). \tag{17}$$

The SMNet can be replaced with any imputation network. In our experiments, we use TimesNet as the SMNet.

## C.2   The inference process of Short-term Imputer

During the inference process, we partition the entire dataset using a sliding window of length $L$ and a sliding step of $L$. Each window is input as $\mathbf{X}_t$ into the Short-term Imputer, which outputs the imputed result $\mathbf{Z}_t$ for the current window. After imputing all the windows, we obtain the final imputed result $\hat{\mathbf{I}}$ from the Short-term Imputer.

# D  Details of the Meta-Learning Module

The Meta-Learning Module (referred to as the Gate) is a pivotal component of our system, tasked with learning the characteristics of the missing data to adaptively allocate weights to the Long-term Imputer and Short-term Imputer outputs. This section offers a comprehensive explanation of the module's architecture, highlighting the design rationale alongside its training and inference processes.

## D.1  The training process

At each training step, a data segment $\mathbf{X}_{tr} \in \mathbb{R}^{(S+L+S) \times C}$ is obtained and divided into three parts: $\mathbf{X}_{tr} = [\mathbf{X}_f, \mathbf{X}_t, \mathbf{X}_b]$. Similar to the Short-term Imputer, the Gate's training process also utilizes only the middle segment, $\mathbf{X}_t \in \mathbb{R}^{L \times C}$. Upon obtaining $\mathbf{X}_t$ and $\mathbf{M}_t$, these matrices are concatenated to form a composite input matrix, $\mathbf{X}_{inp} \in \mathbb{R}^{2L \times C}$. This input is then passed through a linear fully connected layer, augmented by the addition of a fully trainable positional encoding matrix. Formally, this process is expressed as:

$$\mathbf{X}_{inp} = [\mathbf{X}_t, \mathbf{M}_t], \quad \mathbf{X}'_{enc} = \mathbf{X}_{inp}\mathbf{W}'_{enc} + \mathbf{B}'_{enc} + \mathbf{E}'_{pos}. \tag{18}$$

Following this, the encoded data $\mathbf{X}'_{enc}$ is processed through a one-dimensional convolutional layer, which effectively extracts local temporal patterns while addressing boundary effects to ensure a robust feature representation across the sequence. The convolution operation can be formally expressed as:

$$\mathbf{X}_{conv} = \text{Conv1D}(\mathbf{X}'_{enc}). \tag{19}$$

The processed data $\mathbf{X}_{conv}$ is then passed through a series of hidden linear layers, enabling the model to learn complex relationships and interactions within the input features. Finally, a Softmax function is applied to map the embeddings to values between 0 and 1, representing the adaptive mixing ratio. The overall operation can be expressed as:

$$\lambda = \text{Softmax}(\text{MLP}(\mathbf{X}_{conv})), \tag{20}$$

where $\lambda$ represents the mixing ratio, it is used to combine the outputs of the Long-term Imputer, $\mathbf{Y}_t$, and the Short-term Imputer, $\mathbf{Z}_t$. The meta-learning module is subsequently updated through backpropagation using the loss function $\mathcal{L}_t$.

## D.2  The inference process

During inference, the Short-term Imputer partitions the data into windows. The corresponding data and missing mask for each window are then provided as input to the Gate, which computes the mixing ratio for that window. Once the inferences from both the Long-term and Short-term Imputers are complete, the mixing ratio is applied to combine their results, yielding the final imputed output.

# E  Experimental Setup and Analysis

## E.1  Dataset descriptions

Table 9 summarizes the details of all datasets used in our experiments. To ensure consistency and comparability, we adopted the data processing methodology from TimesNet. Each dataset was partitioned into training, validation, and test sets in a 7:1:2 ratio. This division allows for robust model evaluation and minimizes potential biases. The final performance metrics were computed on the test set, which serves as an unseen dataset for assessing the generalization capabilities of the model.

Table 9: Details of the Datasets.

| dataset | Electricity | Traffic | Metr-la | GuangZhou | PEMS04 |
|---------|-------------|---------|---------|-----------|--------|
| channel | 321 | 862 | 207 | 214 | 307 |
| time step | 26304 | 17544 | 34272 | 8784 | 16992 |

## E.2 Showcase results

Figure 8 illustrates the imputation performance of our LSTI, compared with TimesNet and Transformer on the Electricity dataset under the Blackout setting with $L_m = 50$ and $R_m = 30\%$. In this case, the data is missing from the 26th point to the end of the window. Notably, under our global Blackout missing setting, the random selection of missing segments can result in overlapping intervals, leading to extended missing segments. This overlap explains why the missing segment in this case exceeds the predefined length of 50.

As illustrated in the figure, the Transformer model performs poorly when dealing with continuous missing data, with its imputation results approximating simple mean imputation. TimesNet demonstrates better performance, with some reconstructed values aligning closely with the ground truth. However, it struggles to capture sharp transitions and maintain consistency in highly dynamic regions. In contrast, LSTI addresses these challenges effectively, reconstructing missing segments with high accuracy while preserving short-term variability and long-term temporal dependencies. This leads to imputation results that closely align with the ground truth, underscoring the robustness and adaptability of LSTI, positioning it as a highly effective solution for time series data imputation, particularly in scenarios involving continuous data loss.

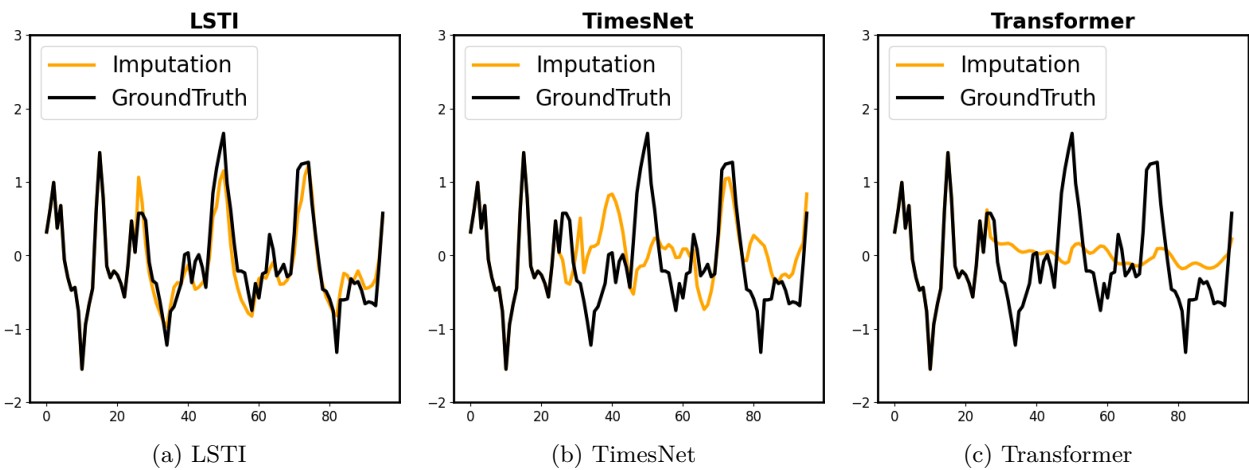

(a) LSTI           (b) TimesNet           (c) Transformer

Figure 8: Visualization of imputation results on the Electricity dataset under the Blackout setting with $L_m = 50$ and $R_m = 30\%$. The black line represents the ground truth, while the orange line indicates the imputed values. The missing segment begins at the 26th point in the sequence.

## E.3 Analysis of computational resource

The experiments in this study were conducted on a high-performance computing system with the following specifications: eight NVIDIA GeForce RTX 4090 GPUs, each equipped with 24GB of VRAM, enabling efficient execution of deep learning and high-performance computational tasks. The system is powered by a 128-core AMD EPYC 7513 processor, which provides substantial parallel processing power. Additionally, the server is configured with 503GB of RAM, facilitating the handling of large datasets and memory-intensive operations.

We comprehensively evaluate the model efficiency across various datasets, assessing metrics such as memory consumption, overall runtime, and imputation performance. Figure 9 presents these results, where the x-axis

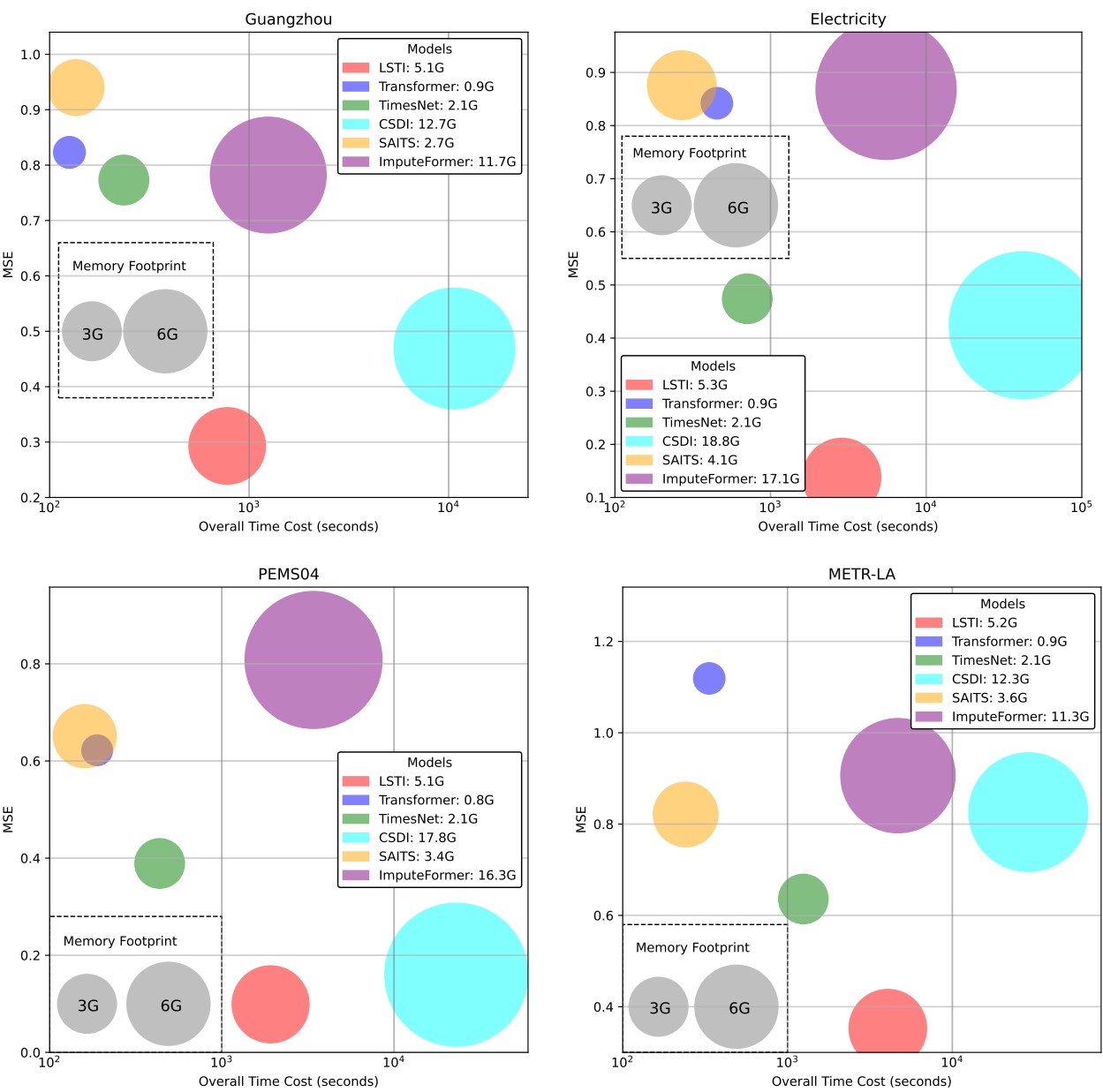

Figure 9: A comprehensive comparison of model efficiency on the Guangzhou, Electricity, PEMS04, and METR-LA datasets. All results are obtained under the settings of $L_m = 10$ and $R_m = 30\%$.

represents the overall runtime of each model, the y-axis indicates the imputation MSE, and the bubble size reflects the memory usage of each model. All experiments are conducted with TimesNet as the backbone, using settings of $L_m = 10$ and $R_m = 30\%$.

The results demonstrate that, although LSTI incurs higher computational overhead compared to some lightweight models, the significant improvement in accuracy more than offsets this cost. Additionally, for resource-intensive models like CSDI, LSTI not only reduces computational demands but also delivers superior performance. By striking a balance between computational efficiency and state-of-the-art accuracy, LSTI establishes itself as a robust and reliable solution for time series data imputation, particularly in addressing datasets with long consecutive missing segments.

# F   Additional experiments on more datasets.

To evaluate LSTI's performance across diverse domains and datasets with varying distributions, we conducted additional experiments on two widely recognized benchmarks: Weather (Wetterstation) and ETTm1 (Zhou et al., 2021). The Weather dataset consists of 10-minute interval data from 2020, encompassing 21 meteorological indicators such as air temperature and humidity. The ETTm1 dataset provides 15-minute interval data, capturing load characteristics of seven oil and power transformers over a two-year period from July 2016 to July 2018. The results of these experiments are summarized in Tables 11 and 12. To ensure a fair and consistent comparison, all experimental settings align with those outlined in the main text.

Table 10: Details of the Datasets.

| dataset | Weather | ETTm1 |
|---|---|---|
| channel | 21 | 7 |
| time step | 52696 | 69680 |

Table 11: Imputation results on real datasets with different missing lengths and a fixed 30% missing rate. The label "_F" indicates models using forecasting methods.

| | $L_m$ | LSTI | | Transformer | | CSDI | | SAITS | | TimesNet | | Imputeformer | | TimesNet_F | | iTransformer_F | |
|---|---|---|---|---|---|---|---|---|---|---|---|---|---|---|---|---|---|
| | | MAE | MSE | MAE | MSE | MAE | MSE | MAE | MSE | MAE | MSE | MAE | MSE | MAE | MSE | MAE | MSE |
| Weather | 10 | 0.120 | **0.071** | 0.388 | 0.298 | **0.104** | 0.143 | 0.411 | 0.342 | 0.154 | 0.098 | 0.269 | 0.194 | 0.298 | 0.264 | 0.283 | 0.215 |
| | 30 | **0.153** | **0.108** | 0.486 | 0.431 | 0.167 | 0.574 | 0.500 | 0.468 | 0.205 | 0.164 | 0.468 | 0.430 | 0.336 | 0.306 | 0.297 | 0.256 |
| | 50 | **0.197** | **0.144** | 0.503 | 0.450 | 0.239 | 0.657 | 0.548 | 0.537 | 0.240 | 0.192 | 0.508 | 0.500 | 0.345 | 0.343 | 0.286 | 0.244 |
| | 100 | **0.295** | **0.239** | 0.546 | 0.505 | 0.367 | 0.759 | 0.641 | 0.708 | 0.321 | 0.278 | 0.564 | 0.562 | 0.386 | 0.360 | 0.365 | 0.332 |
| | 300 | **0.413** | **0.358** | 0.546 | 0.512 | 0.512 | 1.030 | 0.675 | 0.757 | 0.449 | 0.409 | 0.556 | 0.538 | 0.486 | 0.464 | 0.486 | 0.466 |
| ETTm1 | 10 | **0.263** | **0.158** | 0.559 | 0.628 | 0.287 | 0.206 | 0.624 | 0.848 | 0.536 | 0.593 | 0.435 | 0.378 | 0.460 | 0.453 | 0.429 | 0.386 |
| | 30 | **0.362** | **0.293** | 0.718 | 0.889 | 0.546 | 0.673 | 0.704 | 1.000 | 0.721 | 1.025 | 0.646 | 0.726 | 0.492 | 0.518 | 0.464 | 0.472 |
| | 50 | **0.430** | **0.418** | 0.716 | 0.900 | 0.637 | 0.832 | 0.716 | 1.023 | 0.718 | 1.051 | 0.721 | 0.898 | 0.495 | 0.534 | 0.471 | 0.483 |
| | 100 | **0.509** | **0.547** | 0.694 | 0.855 | 0.747 | 1.073 | 0.695 | 0.930 | 0.774 | 1.114 | 0.699 | 0.864 | 0.546 | 0.623 | 0.513 | 0.551 |
| | 300 | **0.578** | **0.632** | 0.769 | 1.023 | 0.976 | 1.609 | 0.777 | 1.123 | 0.733 | 1.124 | 0.778 | 1.045 | 0.705 | 0.917 | 0.687 | 0.880 |

Table 12: Imputation results on real datasets with different missing rates and a fixed 50 missing length. The label "_F" indicates imputation using forecasting methods.

| | $R_m$ | LSTI | | Transformer | | CSDI | | SAITS | | TimesNet | | Imputeformer | | TimesNet_F | | iTransformer_F | |
|---|---|---|---|---|---|---|---|---|---|---|---|---|---|---|---|---|---|
| | | MAE | MSE | MAE | MSE | MAE | MSE | MAE | MSE | MAE | MSE | MAE | MSE | MAE | MSE | MAE | MSE |
| Weather | 10% | **0.183** | **0.134** | 0.512 | 0.492 | 0.243 | 0.274 | 0.443 | 0.402 | 0.220 | 0.175 | 0.483 | 0.465 | 0.256 | 0.217 | 0.235 | 0.177 |
| | 30% | **0.196** | **0.143** | 0.503 | 0.450 | 0.206 | 0.375 | 0.548 | 0.537 | 0.240 | 0.192 | 0.508 | 0.500 | 0.304 | 0.269 | 0.286 | 0.244 |
| | 50% | **0.213** | **0.154** | 0.509 | 0.460 | 0.288 | 0.715 | 0.600 | 0.622 | 0.259 | 0.224 | 0.540 | 0.536 | 0.430 | 0.441 | 0.395 | 0.358 |
| | 70% | **0.256** | **0.192** | 0.425 | 0.372 | 0.333 | 1.221 | 0.600 | 0.652 | 0.301 | 0.273 | 0.538 | 0.538 | 0.490 | 0.502 | 0.479 | 0.453 |
| ETTm1 | 10% | 0.428 | 0.409 | 0.677 | 0.822 | 0.550 | 0.668 | 0.677 | 0.931 | 0.718 | 1.051 | 0.640 | 0.749 | 0.434 | 0.424 | **0.395** | **0.363** |
| | 30% | **0.439** | **0.412** | 0.716 | 0.900 | 0.677 | 0.928 | 0.716 | 1.023 | 0.722 | 1.099 | 0.721 | 0.898 | 0.499 | 0.539 | 0.471 | 0.483 |
| | 50% | **0.454** | **0.424** | 0.687 | 0.899 | 0.693 | 0.985 | 0.737 | 1.095 | 0.761 | 1.129 | 0.733 | 0.954 | 0.556 | 0.638 | 0.541 | 0.606 |
| | 70% | **0.469** | **0.471** | 0.640 | 0.797 | 0.741 | 1.061 | 0.743 | 1.085 | 0.739 | 1.128 | 0.731 | 0.946 | 0.623 | 0.744 | 0.607 | 0.711 |

As illustrated in the tables, LSTI consistently outperforms baseline methods across the majority of experimental settings, with particularly notable gains under conditions of high missing rates and long consecutive missing segments. Specifically, LSTI achieves an average improvement of 49.25% on the Weather dataset and 52.67%

on the ETTm1 dataset. These substantial enhancements underscore LSTI's strong imputation capabilities in scenarios characterized by challenging data gaps. Furthermore, the results highlight LSTI's remarkable generalizability across diverse datasets with varying characteristics, solidifying its position as a robust and effective solution for addressing data imputation challenges in complex real-world applications.

## G Baseline Comparison and Analysis: SSSD

In this section, we analyze the SSSD method (Alcaraz & Strodthoff, 2022), which has been reported to outperform baselines such as CSDI and SAITS. Additionally, SSSD reported evaluation results under the Blackout missing pattern in its original paper. To further investigate its performance, we conducted additional experiments comparing LSTI with SSSD and examined why SSSD struggles to effectively address the global Blackout missing pattern configured in our experiments.

### G.1 Comparison between LSTI and SSSD

To further evaluate the SSSD method, we conducted additional experiments following the experimental settings outlined in the main text. All hyperparameters for SSSD were set to their default values as specified in its original implementation. To ensure a fair and consistent comparison, we applied the standardized data preprocessing method used in this study to all datasets. It is worth noting that SSSD employs dataset-specific preprocessing strategies in its original experiments, which could potentially influence its performance outcomes.

The experimental results are summarized in Tables 13 and 14. Across all datasets and experimental configurations, SSSD consistently underperformed when compared to our proposed LSTI method. This performance gap was particularly pronounced on datasets such as Traffic and Electricity, where SSSD exhibited significantly suboptimal results. These findings not only underscore the robustness and adaptability of LSTI but also highlight the challenges faced by SSSD in effectively handling the global Blackout missing pattern adopted in our experiments.

Table 13: Comparison of imputation results between LSTI and SSSD across different datasets under a fixed 30% missing rate.

| | | Electricity | | Traffic | | Metr-la | | GuangZhou | | PEMS04 | |
|---|---|---|---|---|---|---|---|---|---|---|---|
| | $L_m$ | MAE | MSE | MAE | MSE | MAE | MSE | MAE | MSE | MAE | MSE |
| LSTI | 10 | **0.249** | **0.138** | **0.269** | **0.296** | **0.342** | **0.353** | **0.367** | **0.293** | **0.213** | **0.099** |
| | 30 | **0.242** | **0.129** | **0.281** | **0.316** | **0.434** | **0.568** | **0.387** | **0.323** | **0.263** | **0.132** |
| | 50 | **0.255** | **0.144** | **0.301** | **0.341** | **0.508** | **0.747** | **0.395** | **0.321** | **0.267** | **0.142** |
| | 100 | **0.278** | **0.171** | **0.334** | **0.381** | **0.478** | **0.714** | **0.420** | **0.350** | **0.425** | **0.328** |
| | 300 | **0.370** | **0.251** | **0.450** | **0.547** | **0.712** | **1.225** | **0.495** | **0.466** | **0.569** | **0.513** |
| SSSD | 10 | 1.308 | 3.667 | 1.450 | 4.435 | 1.046 | 2.635 | 0.739 | 1.290 | 1.132 | 2.734 |
| | 30 | 1.309 | 3.672 | 1.455 | 4.465 | 1.057 | 2.664 | 0.757 | 1.345 | 1.245 | 3.425 |
| | 50 | 1.308 | 3.666 | 1.454 | 4.457 | 1.157 | 3.028 | 0.993 | 2.252 | 1.250 | 3.441 |
| | 100 | 1.310 | 3.676 | 1.458 | 4.487 | 1.233 | 3.283 | 1.149 | 2.923 | 1.248 | 3.444 |
| | 300 | 1.307 | 3.663 | 1.462 | 4.507 | 1.239 | 3.316 | 1.147 | 2.913 | 1.255 | 3.467 |

### G.2 Analysis of SSSD

In the following, we analyze the factors leading to SSSD's suboptimal performance from several perspectives.

Table 14: Comparison of imputation results between LSTI and SSSD across different datasets under a fixed 50 missing length.

| | $R_m$ | Electricity | | Traffic | | Metr-la | | GuangZhou | | PEMS04 | |
|---|---|---|---|---|---|---|---|---|---|---|---|
| | | MAE | MSE | MAE | MSE | MAE | MSE | MAE | MSE | MAE | MSE |
| LSTI | 10% | **0.224** | **0.108** | **0.265** | **0.308** | **0.478** | **0.710** | **0.410** | **0.355** | **0.232** | **0.109** |
| | 30% | **0.255** | **0.144** | **0.301** | **0.341** | **0.508** | **0.747** | **0.395** | **0.321** | **0.267** | **0.142** |
| | 50% | **0.269** | **0.157** | **0.292** | **0.322** | **0.568** | **0.867** | **0.428** | **0.368** | **0.349** | **0.235** |
| | 70% | **0.321** | **0.209** | **0.366** | **0.428** | **0.621** | **1.025** | **0.491** | **0.457** | **0.478** | **0.388** |
| SSSD | 10% | 1.186 | 3.055 | 1.432 | 4.335 | 0.980 | 2.357 | 0.531 | 0.768 | 1.032 | 2.601 |
| | 30% | 1.255 | 3.446 | 1.454 | 4.457 | 1.057 | 2.664 | 0.757 | 1.345 | 1.182 | 3.135 |
| | 50% | 1.256 | 3.449 | 1.451 | 4.432 | 1.134 | 2.879 | 1.044 | 2.450 | 1.192 | 3.176 |
| | 70% | 1.309 | 3.672 | 1.455 | 4.460 | 1.225 | 3.247 | 1.150 | 2.927 | 1.250 | 3.447 |

First, while SSSD reports evaluation results under the Blackout missing pattern in its original paper, the implementation in its publicly available code adopts a simplified and less realistic approach. Specifically, SSSD randomly masks a fixed-length segment within each imputation window, ensuring uniform missing ratios and segment lengths across all windows. This significantly reduces the complexity of the missing data patterns, making the model's training and inference processes easier. In contrast, the global Blackout missing pattern addressed in this study better reflects real-world scenarios, where missing segments are randomly distributed across the entire dataset rather than confined to specific imputation windows. This introduces variability in missing ratios and continuous missing segment lengths across different windows, substantially increasing the difficulty of the imputation tasks.

Second, the original SSSD paper reports experimental results under the Blackout missing pattern only for the PTB-XL dataset, while results for other datasets, such as MuJoCo and Electricity, are limited to the random missing pattern. This omission raises concerns about the generalizability of the SSSD model. As evidenced by the results in Tables 13 and 14, as well as additional experiments, SSSD demonstrates limited effectiveness on larger datasets, particularly under the more complex global Blackout missing pattern. These findings underscore its challenges in addressing more demanding data imputation scenarios.

Third, SSSD employs an identical missing pattern during both training and testing. In other words, during training, the model is provided with prior knowledge of the fixed missing ratio and continuous missing length for each training window. However, such prior knowledge is often impractical to obtain in real-world applications. While this approach allows SSSD to perform exceptionally well under a predefined fixed missing pattern, its effectiveness diminishes when faced with more complex missing scenarios, such as varying missing ratios and continuous missing lengths across different windows.

In summary, our proposed method, LSTI, demonstrates significant advantages over SSSD, particularly when evaluated under the global Blackout missing pattern. This highlights its robustness and adaptability to complex, real-world conditions.

# H  Discussion on Implicit Neural Representations (INR)

Implicit Neural Representations (INRs) constitute a class of neural networks designed to parameterize continuous functions by mapping input coordinates, such as temporal indices, to corresponding output values. As an emerging area in time series analysis, INRs represent data as continuous functions rather than discrete samples, facilitating the modeling of temporal dynamics in a coordinate-based manner. This framework is particularly effective for applications involving irregularly sampled data, interpolation, or the reconstruction of missing values, making it well-suited for addressing the global Blackout missing pattern discussed in this paper.

## H.1  Comparison between LSTI and TimeFlow

In this section, we present TimeFlow  (Naour et al., 2023) as an additional baseline for comparison with LSTI and provide a detailed discussion of the strengths and limitations of INR-based methods in addressing the challenges posed by the global Blackout missing data pattern.

Table 15: Comparison of imputation results between LSTI and TimeFlow across different datasets under a fixed 30% missing rate

|  | $L_m$ | Electricity | | Traffic | | Metr-la | | GuangZhou | | PEMS04 | |
|---|---|---|---|---|---|---|---|---|---|---|---|
|  |  | MAE | MSE | MAE | MSE | MAE | MSE | MAE | MSE | MAE | MSE |
| LSTI | 10 | **0.249** | **0.138** | **0.269** | **0.296** | 0.342 | 0.353 | **0.367** | **0.293** | **0.213** | **0.099** |
|  | 30 | **0.242** | **0.129** | **0.281** | **0.316** | 0.434 | 0.568 | **0.387** | **0.323** | 0.263 | 0.132 |
|  | 50 | **0.255** | **0.144** | **0.301** | **0.341** | 0.508 | 0.747 | **0.395** | **0.321** | **0.267** | **0.142** |
|  | 100 | **0.278** | **0.171** | **0.334** | **0.381** | 0.478 | 0.714 | **0.420** | **0.350** | **0.425** | **0.328** |
|  | 300 | **0.370** | **0.251** | **0.450** | **0.547** | **0.712** | **1.225** | **0.495** | **0.466** | **0.569** | **0.513** |
| TimeFlow | 10 | 0.811 | 0.984 | 0.744 | 1.106 | **0.261** | **0.277** | 0.441 | 0.378 | 0.226 | 0.106 |
|  | 30 | 0.884 | 1.134 | 0.941 | 1.491 | **0.383** | **0.417** | 0.529 | 0.528 | **0.238** | **0.122** |
|  | 50 | 0.934 | 1.397 | 1.228 | 2.353 | 0.699 | 1.202 | 0.709 | 0.933 | 0.309 | 0.255 |
|  | 100 | 0.972 | 1.514 | 1.477 | 3.389 | 0.892 | 1.909 | 1.081 | 2.097 | 0.580 | 0.630 |
|  | 300 | 1.484 | 4.467 | 1.787 | 5.476 | 1.142 | 3.387 | 1.114 | 2.371 | 1.292 | 2.730 |

Tables 15 and 16 summarize the performance of TimeFlow on our datasets under the experimental settings. The results indicate that LSTI consistently outperforms TimeFlow across most scenarios, reaffirming LSTI as the state-of-the-art solution for addressing the challenges posed by the Blackout missing pattern. Specifically, LSTI demonstrates superior robustness and adaptability, effectively handling variations in missing rates and continuous missing lengths.

It is important to note that TimeFlow is not suitable for imputing large-scale datasets. To ensure a fair comparison, we adopted the same training and testing split as LSTI and used the full set of observed data points for training. Under this setup, TimeFlow requires significant computational resources, taking considerable time and memory to complete a single training run. This is corroborated by the original TimeFlow paper, where the datasets used were sub-samples of the original data, each containing fewer than 2000 data points. When we trained TimeFlow using a sub-sample of similar size under the Blackout missing pattern, the results were highly unsatisfactory, with some cases yielding an MSE greater than 1000. Therefore, TimeFlow has notable limitations when applied to large-scale datasets.

It can be observed that TimeFlow exhibits commendable performance on specific datasets when the missing rates are lower, and the lengths of continuous missing segments are relatively short. However, on other datasets, TimeFlow struggles significantly, showing subpar results even at the low-missing-rate settings.

Table 16: Comparison of imputation results between LSTI and TimeFlow across different datasets under a fixed 50 missing length.

| | $R_m$ | Electricity | | Traffic | | Metr-la | | GuangZhou | | PEMS04 | |
|---|---|---|---|---|---|---|---|---|---|---|---|
| | | MAE | MSE | MAE | MSE | MAE | MSE | MAE | MSE | MAE | MSE |
| LSTI | 10% | **0.224** | **0.108** | **0.265** | **0.308** | 0.478 | 0.710 | **0.410** | **0.355** | **0.232** | **0.109** |
| | 30% | **0.255** | **0.144** | **0.301** | **0.341** | **0.508** | **0.747** | **0.395** | **0.321** | **0.267** | **0.142** |
| | 50% | **0.269** | **0.157** | **0.292** | **0.322** | **0.568** | **0.867** | **0.428** | **0.368** | **0.349** | **0.235** |
| | 70% | **0.321** | **0.209** | **0.366** | **0.428** | **0.621** | **1.025** | **0.491** | **0.457** | **0.478** | **0.388** |
| TimeFlow | 10% | 0.834 | 1.020 | 0.918 | 1.591 | **0.469** | **0.538** | 0.622 | 0.678 | 0.307 | 0.167 |
| | 30% | 0.934 | 1.397 | 1.228 | 2.353 | 0.624 | 1.131 | 0.709 | 0.933 | 0.309 | 0.255 |
| | 50% | 1.133 | 2.114 | 1.487 | 3.614 | 0.699 | 1.202 | 0.728 | 0.943 | 0.389 | 0.299 |
| | 70% | 1.356 | 3.159 | 1.635 | 4.460 | 1.031 | 2.486 | 1.452 | 4.475 | 0.847 | 1.639 |

As the missing rate and continuous missing length increase, TimeFlow's performance deteriorates sharply, sometimes leading to severe deviations that render the imputation results unreliable.

These observations underscore the strengths and limitations of INR-based methods like TimeFlow. While they offer distinct advantages over traditional imputation techniques, especially for structured missing data, their effectiveness is not universal. The results suggest that INR-based methods require careful calibration and may not always generalize well to more complex or diverse missing data scenarios. Consequently, practitioners should exercise caution when adopting such methods.

## H.2 Analysis of INR-based methods

The primary distinction between INR-based methods and traditional deep learning imputation techniques is how they conceptualize time series data. Traditional methods treat time series as discrete data points, while INR-based approaches model time series as continuous functions, with each data point associated with a specific time coordinate. In this section, we examine the factors contributing to the strong performance of INR-based methods, exemplified by TimeFlow, on specific datasets and missing rates, as well as the reasons for their poor performance under other conditions.

First, traditional deep learning methods often impute missing data by replacing missing values with zeros in imputation windows, relying on the network to infer missing information. However, this approach struggles with real-world scenarios where varying missing rates and patterns across windows introduce inconsistencies, degrading performance. INR, in contrast, employs a coordinate-based representation, mapping data to temporal coordinates. This eliminates imputation windows and avoids zero placeholders, enabling more effective handling of missing data. Similarly, LSTI's bidirectional autoregressive imputation strategy removes missing values during regression, ensuring the model receives complete data and mitigating issues caused by varying missing rates, thereby enhancing robustness and generalization.

Secondly, the ability to model continuous functions provides INR-based methods with an advantage in handling long segments of continuous missing data. However, this approach also has limitations, particularly in terms of generalization across diverse datasets. Since INRs rely on deep learning networks to model implicit function fitting, the resulting function is strongly influenced by the network architecture. In other words, a specific INR structure may excel in fitting certain data distributions, but it may struggle significantly with others. For instance, as shown in Tables 15 and 16, TimeFlow performs well on datasets like PEMS04 and Metr-la, but yields poor results on datasets such as Electricity and Traffic. This underscores the selective nature of INR methods and their limited generalization ability across varying datasets.

Third, INR methods perform well in imputing MCAR (Missing Completely at Random) data even under high missing rates but struggle with Blackout missing data under similar conditions. Table 17 shows the results of TimeFlow on the Guangzhou dataset with an MCAR missing pattern, demonstrating a notable performance improvement without a sharp decline as the missing rate increases. This underscores the strengths of INR-based methods in handling high-missing-rate MCAR patterns. However, their performance under the Blackout missing pattern remains suboptimal, with a substantial gap compared to LSTI.

Table 17: Imputation results of TimeFlow on the GuangZhou Dataset under different missing patterns

| $R_m$ | 10% | | 30% | | 50% | | 70% | |
|---|---|---|---|---|---|---|---|---|
| | MAE | MSE | MAE | MSE | MAE | MSE | MAE | MSE |
| MCAR | 0.364 | 0.325 | 0.336 | 0.258 | 0.322 | 0.260 | 0.359 | 0.293 |
| Blackout | 0.622 | 0.678 | 0.709 | 0.933 | 0.728 | 0.943 | 1.452 | 4.475 |

In summary, while INR-based methods exhibit strengths in certain scenarios, LSTI clearly outperforms them when dealing with large-scale Blackout missing patterns, making it the more effective choice in such cases.

## I  Broader Impact

The LSTI presents significant potential in healthcare, finance, and environmental science by addressing missing data in time series. In healthcare, it can enhance patient monitoring and personalized care by accurately imputing gaps in vital signs data. However, there are ethical concerns regarding bias, particularly in vulnerable populations, that may affect clinical outcomes. In finance, LSTI could improve predictive models for market behavior, yet imputed data must be rigorously validated to prevent financial instability and market manipulation. In environmental science, the method can support more accurate climate models, but improper imputation may lead to flawed policy decisions. In summary, across all domains, transparency, model validation, and ethical oversight are critical to ensuring fairness and mitigating potential risks.

