# OpenReview forum: "Long Short-Term Imputer: Handling Consecutive Missing Values in Time Series"
_TMLR — Accepted by TMLR_

### Review · Reviewer_5gmT · 2024-12-14

**Summary Of Contributions:**

This paper introduces the Long Short-Term Imputer (LTSI) model for time series imputation. The LTSI model is specifically designed to effectively handle missing values with a ‘blackout’ pattern, where all channels are missing simultaneously for consecutive timestamps. The model relies on three key components:

- **Long-Term Imputer**: Leverages and combines forward and backward neural predictors.
- **Short-Term Imputer**: Focuses on local missing values.
- **Meta-Weighing Module**: Takes the data and missing mask as input and outputs a scalar weight to combine the long- and short-term imputers.

Experiments on five real-world datasets demonstrate that the LTSI model achieves strong performance compared to the considered baselines.

**Audience:**

Yes

**Claims And Evidence:**

No

**Requested Changes:**

Please see weaknesses section.

**Strengths And Weaknesses:**

**Strengths**

- **S1.** The blackout missing values setting is particularly relevant in practice and challenging to address, making the chosen problem both important and interesting.
- **S2.** The Long Short-Term Imputer model is well-explained and thoughtfully designed. I found the use of the scalar weighting mechanism, which adjusts the emphasis on FPNet or BPNet based on the location of the missing values, to be intuitive and elegant. It was surprising that this part of the model (the long term imputer) does not rely solely on the $\mathcal{L}_p$ term.
- **S3.** The experimental results are impressive, with significant improvements over the considered baselines.


**Weaknesses**

-  **W1.**  The paper is hard to follow, particularly in Section 3 (Method) and Section 4 (Experiments). Specific examples include:
    - Certain aspects of the problem formulation, such as the introduction of the Q matrices, feel unnecessary and overcomplicate the explanations without contributing much to understanding.
    - In the Long-Term Imputer, four terms are used in the loss function, yet no ablation study is provided. As mentioned earlier, it is surprising that the network is not trained solely using the $\mathcal{L}_p$ term. This complexity in the loss terms might hinder reader comprehension.
    - The Short-Term Imputer (SMNet) is insufficiently discussed. Beyond being described as a single self-mapping network, there are no details provided. An explanation or a figure (even in the appendix) would improve clarity.
    - The experimental setup is unclear. For instance, two hyperparameters are mentioned for generating missing data: the overall missing rate ($\mathcal{R}_m$) and the missing length ($\mathcal{L}_m$) of the window. Does this mean that, for example, in Table 1, first row (electricity dataset), you used windows of length 10 with blocks of 30% missing values (i.e., three consecutive values)? Additionally, there seems to be a typo in Table 1, where $\mathcal{R}_m$ is indicated, but $\mathcal{L}_m$ is likely intended.
    - Including figures to clarify the experimental setup would greatly enhance understanding.
-  **W2.** While the results are promising, there are no visualizations to evaluate the quality of imputations compared to the baselines. Adding a few imputation plots would significantly strengthen the paper.
-  **W3.** The paper does not discuss Implicit Neural Representations for time series imputation [1, 2, 3], which is an emerging area in the literature. These models show strong performance and appear particularly suited for blackout scenarios. Including a discussion (and some experiments, if the authors have time) on this topic would add valuable context.


[1] Time Series Continuous Modeling for Imputation and Forecasting with Implicit Neural Representations, Le Naour et al., TMLR 2024

[2] ImputeINR: Enhancing Time Series Imputation with Adaptive Group-based Implicit Neural Representations, Li et al. (https://openreview.net/forum?id=xcPN6Or88c)

[3] MADS: Modulated Auto-Decoding SIREN for time series imputation, Bamford et al, (https://arxiv.org/abs/2307.00868)

---

> ### Author Response · Authors · 2025-01-18
>
> # Response to Reviewer 5gmT
> We thank Reviewer 5gmT for providing a meaningful review and insightful suggestions. We have submitted a revised manuscript addressing many of the questions and suggestions raised during the discussion phase. The content in all appendix sections is newly added, and all modifications in the main text are highlighted in red.
>
> ### **W1.1**:
>
> We streamline the method description by replacing the Q matrix with an indicator function. These modifications are detailed at the beginning of **Section 3**.
>
> ### **W1.2**:
>
> We conduct ablation studies on all the loss functions employed in the Long-Term Imputer. Detailed results and analyses are provided in **Appendix B**. In summary, each loss function proves to be effective, and the combined use of all loss functions consistently outperforms using any single one.
>
> ### **W1.3**:
>
> We include a detailed description of the Short-Term Imputer in **Appendix C**.
>
> ### **W1.4**:
>
> We apologize for the confusion caused. In our experiments, $R_m$ refers to the missing rate, and $L_m$ refers to the length of continuous missing segment. To achieve data with $R_m = 30\%$ and $L_m = 50$ (as described in Table 1), we randomly select intervals of length 50 from the dataset to mask until the overall missing rate reaches 30%. A figure and a detailed explanation have been added in **Section 4.1**.
>
> Additionally, we acknowledge a typo in **Table 1**. We sincerely apologize for this oversight, and the error has been corrected in the latest version.
>
> ### **W1.5**:
>
> We add a figure in **Section 4.1**, providing a detailed description of our experimental setup.
>
> ### **W2**:
>
> We provide a visual showcase of the results in **Appendix E.2**.
>
> ### **W3**:
>
> We present additional experimental results and analysis of INR-based methods in **Appendix H**.
>
> In **Appendix H.1**, we conduct supplementary experiments with the TimeFlow method on our datasets. The results demonstrate that LSTI consistently outperforms TimeFlow in most cases, with particularly noticeable improvements under high missing rates and long continuous missing segments.
>
> In **Appendix H.2**, we provide a detailed analysis of the advantages and limitations of INR-based imputation methods. In summary, INR methods are more effective for imputing high-missing-rate data under the MCAR setting, while LSTI remains the more reliable choice for Blackout missing patterns.

---

> ### Author Response · Authors · 2025-02-06
> **Kind reminder regarding your feedback**
>
> Dear Reviewer 5gmT,
>
> We hope this message finds you well. We understand this might be a particularly busy time for you, and we genuinely appreciate the time you dedicate to reviewing. All the requested changes have been incorporated into the revised version, and further details can be found in our rebuttal.
>
> As we approach the final stages of the review process, we would kindly remind you to review our rebuttal at your earliest convenience. Your feedback is extremely valuable to us, and we would greatly appreciate any additional insights you could provide to help refine and strengthen our work. Should any aspects require further clarification, please do not hesitate to reach out.
>
> Thank you once again for your time, effort, and invaluable contribution to the review process. We eagerly look forward to your feedback.
>
> Best regards,
>
> Authors

---

> > ### Comment · Reviewer_5gmT · 2025-02-08
> > **Answer**
> >
> > I would like to thank the authors for their detailed responses and the additional experiments. In particular, I believe the revisions have improved the clarity of the paper, and the experiments with new baselines strengthen the robustness of the results. Some results and plots seem somewhat surprising to me, but this is likely due to the specific experimental setting.
> >
> > I confirm that I have all the necessary information to make my recommendation.

---

> > > ### Author Response · Authors · 2025-02-08
> > >
> > > Thank you for your thoughtful review and valuable feedback. Your insights are highly appreciated and will be beneficial for our future work.

---

### Review · Reviewer_WCjL · 2024-12-18

**Summary Of Contributions:**

title: Long Short-Term Imputer: Handling Consecutive Missing Values in Time Series

summary: this paper addresses the challenge of imputing missing values in time series data, especially focusing on long-interval consecutive missing values. The authors introduce the Long Short-Term Imputer (LSTI), framework that combines two specialized models: a Long-term Imputer and a Short-term Imputer, along with a Meta-weighting module for adaptive integration. The Long-term Imputer employs bidirectional autoregressive prediction with a consistency regularization loss, designed to capture long-term temporal dependencies. Meanwhile, the Short-term Imputer uses a self-mapping network to handle short-term dependencies effectively. The Meta-weighting module adaptively blends the strengths of both models based on the nature of missing data.

**Audience:**

Yes

**Broader Impact Concerns:**

The statement could improve clarity on why the method is not directly applicable to real-world tasks and how this limitation impacts its utility.

The authors fail to explore potential risks, such as biases in imputation or over-reliance on imputed data. See lack of confidence intervals in review.

**Claims And Evidence:**

Yes

**Requested Changes:**

See weaknesses.

**Strengths And Weaknesses:**

Strengths:
Overall, the method handle the imputation of not just short-interval but also long-interval missing values with what it seems (good) reasonable performance.


Weaknesses:

1) lack of comparison with state-of-the-art models for long term time series imputations, e.g. SSSD https://openreview.net/pdf?id=hHiIbk7ApW which certainly overcome CSDI, SAITS and other baselines. I would suggest at least a proper citation and justification on why not included in the analysis.

2) the definition of missingness scenarios is certainly clear however not consistent with previous works, e.g. SSSD uses completely at random (Random missing), and consecutive missing interval (blackout missing), but the mentioned baseline also investigate random block missing.
I could suggest the authors mention/justify from which literature is that they bring the missigness scenarios definition.

3) lack of computational complexity results. I suggest the authors provide a short section on the number of model parameters, size, and the actual training and generation times for a different datasets of varying lengths and number of channels.

4) lack of confidence intervals. I suggest the authors also provide if not for all experiments for a subset of them confidence intervals e.g. 95% CI. across 100 or so imputations.

5) an extension on the previous point, it would be really interesting to know at how many imputations the variance of the error converges, e.g. for a specific length and number of channels, how many imputations would be ideal to make to get a converged mean.

---

> ### Author Response · Authors · 2025-01-18
>
> # Response to Reviewer WCjL
> Many thanks to Reviewer WCjL for providing thorough insightful comments. We have submitted a revised manuscript addressing many of the questions and suggestions raised during the discussion phase. The content in all appendix sections is newly added, and all modifications in the main text are highlighted in red.
>
> ### **W1**:
>
> We conduct additional experiments with SSSD under our experimental settings, and the results can be found in **Appendix G.1**. Overall, LSTI outperforms SSSD under the Blackout missing pattern.
>
> We would like to clarify that it is not accurate to say that SSSD "certainly overcome CSDI, SAITS, and other baselines." In fact, the original SSSD paper highlights several cases where its performance does not surpass that of CSDI. A detailed analysis of the factors contributing to SSSD’s suboptimal results can be found in **Appendix G.2**.
>
> ### **W2**:
>
> The Blackout missing pattern we used is adapted from the methodology described in [1]. To clarify its differences from the Blackout pattern used by SSSD, we include a figure in **Section 4.1** of the main text. Additionally, a detailed analysis comparing these two patterns is provided in **Section 4.1** and **Appendix G.2**.
>
> Overall, our design leverages a more realistic missing pattern, highlighting LSTI's ability to address complex and practical data scenarios with greater effectiveness.
>
> [1] Khayati, Mourad, Alberto Lerner, Zakhar Tymchenko, and Philippe Cudré-Mauroux. "Mind the gap." Proceedings of the VLDB Endowment 13 (2020): 768-782.
>
> ### **W3**:
>
> In **Appendix E.3**, we provide detailed computational resources and a comprehensive comparison figure of model efficiency. Overall, while LSTI's computational overhead is moderate, its substantial improvement in accuracy far surpasses that of other methods, making it acceptable.
>
> ### **W4**:
>
> We provide the 95% confidence intervals for all experimental results in **Appendix A.2**.
>
> ### **W5**:
>
> We conduct 30 repeated experiments to examine the convergence of LSTI imputation results. The detailed results are provided in **Appendix A.3**, where convergence is typically achieved after around 5 repetitions.

---

> ### Author Response · Authors · 2025-02-06
> **Kind reminder regarding your feedback**
>
> Dear Reviewer WCjL,
>
> We hope this message finds you well. We understand this might be a particularly busy time for you, and we genuinely appreciate the time you dedicate to reviewing. All the requested changes have been incorporated into the revised version, and further details can be found in our rebuttal.
>
> As we approach the final stages of the review process, we would kindly remind you to review our rebuttal at your earliest convenience. Your feedback is extremely valuable to us, and we would greatly appreciate any additional insights you could provide to help refine and strengthen our work. Should any aspects require further clarification, please do not hesitate to reach out.
>
> Thank you once again for your time, effort, and invaluable contribution to the review process. We eagerly look forward to your feedback.
>
> Best regards,
>
> Authors

---

### Review · Reviewer_bUFZ · 2025-01-08

**Summary Of Contributions:**

The submission introduces valuable contributions to time series imputation, particularly focusing on the challenge of consecutive missing values. The proposed Long Short-term Imputer (LSTI) framework is a notable highlight, combining long-term and short-term imputation strategies to address varying patterns of missing data. By adaptively weighting the outputs of both imputers, LSTI effectively improves the quality of imputation. A key strength of this framework is the Meta-weighting module, which dynamically adjusts the imputation strategy based on input data characteristics. This innovation proves especially beneficial for handling long-interval consecutive missing data. The authors validate their approach through experiments on five real-world datasets, showing that LSTI consistently outperforms existing imputation and forecasting methods, making a strong case for its practical utility. Additionally, the framework’s end-to-end design streamlines the imputation process, reducing complexity compared to traditional methods. This makes the approach more accessible and efficient for real-world applications. Overall, the submission offers a robust and adaptable solution for time series imputation, filling a critical gap in current methodologies while demonstrating its effectiveness through thorough experimentation!

**Audience:**

Yes

**Broader Impact Concerns:**

In my view, the authors could strengthen their Broader Impact Statement by delving deeper into the ethical considerations associated with their work. The proposed Long Short-Term Imputer (LSTI) demonstrates great potential for addressing consecutive missing values in time series data, with applications spanning critical areas like healthcare, finance, and environmental monitoring. While the current statement mentions the absence of obvious negative social impacts, this assessment might oversimplify the broader ethical implications tied to advancements in time series imputation. A more nuanced discussion could enhance the paper’s perspective on its societal and ethical dimensions.

A critical point for further exploration is the potential application of this technology in sensitive areas where data privacy and accuracy are crucial. For example, in healthcare, more precise imputation of missing patient data could significantly enhance diagnostic accuracy and treatment outcomes. However, this also highlights concerns about the reliability of imputed data and the risks of making critical medical decisions based on reconstructed information. It would be valuable for the authors to discuss how their method can be validated and constrained to prioritize patient safety and uphold ethical standards in such high-stakes environments. Similarly, the financial sector could see notable benefits, such as improved market predictions and risk management. Yet, this raises important issues, including the potential for market manipulation and unfair competitive advantages. The authors are encouraged to address safeguards or regulatory frameworks that could mitigate these risks, ensuring that the method does not unintentionally deepen existing inequalities or create new ethical dilemmas in financial markets.

An important consideration that seems to be missing from the discussion is the environmental impact of the LSTI model. While the paper does an excellent job of highlighting the model's performance, it overlooks the computational resources required for training and deploying the model at scale. With growing concerns about the carbon footprint associated with AI technologies, it would be valuable for the authors to address the energy efficiency of their approach in comparison to existing methods. Additionally, exploring strategies to reduce the model's environmental impact would strengthen the overall contribution and align it with the broader goals of sustainable AI.

The authors should also consider discussing the potential risks associated with their method being misused in scenarios like data manipulation or falsification. While the primary goal is to accurately impute missing data, there’s an inherent risk that such techniques could be misapplied to create misleading or fraudulent datasets. Including a thoughtful discussion on how the approach could be misused, along with strategies to mitigate these risks, would add an important layer of ethical consideration to the paper.

In closing, I’d like to highlight the importance of considering the broader societal implications of improved time series imputation. For example, it would be valuable to explore how this technology could influence decision-making in areas like urban planning, climate change modeling, or economic forecasting. Additionally, reflecting on the potential societal benefits and risks could provide a more well-rounded perspective on its broader impact.

**Claims And Evidence:**

Yes

**Requested Changes:**

Based on my review, I would propose several adjustments that would strengthen the work and potentially secure a recommendation for acceptance at TMLR. These adjustments range from critical improvements to suggestions that would enhance the overall quality and impact of the research. A critical adjustment that I believe is a more comprehensive analysis of the computational complexity and training time of the LSTI model compared to existing methods. While the work demonstrates impressive performance improvements, it lacks a detailed discussion of the computational resources required to achieve these results. This information is crucial for assessing the practical applicability of the proposed method in real-world scenarios. The authors should provide benchmarks comparing the training and inference times of LSTI with baseline models across different datasets and missing data patterns.

A key area for improvement would be to broaden the scope of the experimental evaluation by including a more diverse set of datasets, especially from domains that are not currently represented in the study. While the choice of datasets used is valid, incorporating data from sectors like healthcare, finance, or environmental monitoring could significantly enhance the paper’s claims about the model's generalizability and its broader applicability. Additionally, providing a deeper analysis of how the model performs on datasets with different characteristics—such as varying seasonality or trend—would further strengthen the work. This would offer valuable insights into the model's flexibility and help pinpoint any potential weaknesses or areas that could be refined in future research.

A valuable addition to the paper would be a more in-depth discussion on the interpretability of the model's decisions, especially in relation to the Meta-weighting module. Visualizations or case studies showcasing how the model balances long-term and short-term dependencies in various scenarios could significantly enhance the reader's understanding of the decision-making process and increase its practical applicability. Although not essential for acceptance, a comparison with more specialized methods tailored to handling long-interval missing data in specific domains could further solidify the paper's contribution. This would position LSTI within the wider field of imputation techniques and highlight its unique strengths over domain-specific approaches.

A useful addition to the work would be a more detailed exploration of how the model performs under various hyperparameter settings. This analysis would help assess the robustness of the method and provide practical guidance for users looking to implement LSTI in different contexts. Additionally, to improve the paper’s impact and support reproducibility, I strongly recommend that the authors release their code and pre-trained models, accompanied by clear documentation on how to use and extend the LSTI framework. This would make it easier for other researchers to build upon the work and promote its application in the wider scientific community.

I have one final comment regarding the ablation study. While the current analysis is certainly informative, I believe it could be further strengthened by incorporating a more detailed breakdown of the contributions of each component. In particular, examining the roles of the consistency loss and the Meta-weighting module in greater depth would offer a clearer understanding of how these elements influence the model’s overall performance. This level of detail would provide valuable insights into the design choices and their impact.

**Strengths And Weaknesses:**

The work presents a well-thought-out approach to tackling the difficult problem of imputing long stretches of consecutive missing values in time series data. This is a common challenge in real-world applications where many existing methods fall short. The proposed Long Short-Term Imputer (LSTI) stands out by combining both long-term and short-term imputation strategies. What I find particularly noteworthy is how the model adaptively balances the contributions of these two components, allowing it to handle datasets with different patterns of missing data effectively. This demonstrates a solid understanding of the complexities involved in time series analysis. By capturing diverse temporal dependencies, LSTI not only improves imputation quality but also offers a practical solution for practitioners working with real-world datasets. Overall, this is a valuable contribution to the field of time series imputation, with the potential for significant real-world impact.

I found one of the standout strengths of this work to be its thoughtful and comprehensive approach to addressing both long-term and short-term dependencies in time series data. The use of the Long-Term Imputer, which incorporates forward and backward prediction networks along with a consistency loss, is particularly effective in capturing long-range temporal relationships. This bidirectional strategy, combined with an autoregressive imputation method, demonstrates a robust ability to handle extended missing intervals, setting it apart from many existing approaches in the field. The addition of a Short-Term Imputer is a thoughtful and practical choice, as it allows the model to effectively capture local temporal patterns. This flexibility enhances the model's adaptability, making it well-suited for handling a range of missing data scenarios, from long consecutive gaps to more sporadic omissions. Such a dual-focused strategy reflects a deep and nuanced understanding of the real-world challenges associated with time series data.

The Meta-weighting module is an impressive and well-thought-out addition to the LSTI architecture. It dynamically adjusts the importance of long-term and short-term dependencies based on the characteristics of the input data, which is a critical capability for handling diverse missing data patterns. This flexibility makes the module particularly effective in challenging scenarios, such as those involving long stretches of consecutive missing values—situations where traditional methods often fall short. The approach demonstrates a thoughtful and forward-looking perspective, highlighting the authors' recognition of the importance of adaptive methodologies in addressing real-world data challenges. This is a meaningful step forward compared to static imputation strategies.

One of the key strengths of this submission is the thorough experimental evaluation. The authors present comprehensive comparisons across five real-world datasets, clearly demonstrating the superior performance of LSTI compared to state-of-the-art imputation and forecasting methods. The substantial reduction in error rates highlights its strong potential for real-world applications. Moreover, the inclusion of ablation studies offers valuable insights into the roles of individual components, effectively justifying the design choices made in the LSTI architecture. This careful and rigorous validation process adds significant weight to the theoretical claims and bolsters the reliability of the proposed approach. The authors' thorough exploration of various missing data patterns, such as MCAR and different types of consecutive missing patterns, demonstrates the robustness and adaptability of their model. By addressing these scenarios, the evaluation effectively highlights the method's potential for practical use in a wide range of real-world applications.

The end-to-end learning framework of LSTI is another notable strength, as it simplifies the imputation process and makes it more accessible for practical use. This is especially important because many practitioners may not have in-depth expertise in deep learning or time series analysis. By offering a more straightforward approach, the authors effectively enhance the usability of their model, which is crucial for promoting its adoption across different fields and ensuring it can be implemented with ease in real-world applications.

The theoretical foundation of the approach is well-established, particularly in how the consistency loss and weighted ensemble strategy for the Long-Term Imputer are formulated. This demonstrates a clear and deep understanding of the core challenges involved in time series imputation.

While this work has several notable strengths, there are a few areas that warrant further consideration. For instance, the computational complexity and training time of the LSTI model, especially in comparison to existing methods, are not explicitly discussed. These factors could be important when evaluating the model's suitability for practical deployment. Additionally, while the model performs well across various datasets, a deeper analysis of its behavior on datasets with differing characteristics—such as varying levels of seasonality or trend—could provide more insight into its generalizability and robustness in diverse real-world scenarios.

One area that could benefit from further development is the explanation of the training process and the specific roles of the different components within the LSTI framework. Although the overall structure is presented, a more in-depth description of how the Long-term Imputer, Short-term Imputer, and Meta-weighting module interact would help clarify the model’s functionality. It would be helpful to provide more detail on the implementation of consistency regularization and its impact on the training dynamics of both the forward and backward prediction networks. This additional information would improve the reader's comprehension of the model's inner workings.

Additionally, the paper would benefit from a more detailed discussion on the limitations of the proposed method, especially in situations where the missing data characteristics differ significantly from those in the training datasets. For example, it would be insightful to examine the model's performance under conditions of non-random missingness or when the temporal patterns in the data are irregular. Highlighting these aspects would offer a more balanced perspective on the model's strengths and potential challenges, which is crucial for shaping future research directions and its real-world applications.

Additionally, the interpretability of the model’s decisions, especially in relation to the Meta-weighting module, could benefit from further elaboration. Offering more detailed explanations of how the model manages the balance between long-term and short-term dependencies in various scenarios would provide deeper insight into its decision-making process. Lastly, while the authors compare their method to several baselines, including recent state-of-the-art models, incorporating comparisons with more specialized techniques tailored for handling long-interval missing data in specific domains would add further value to the paper’s contribution.

---

> ### Author Response · Authors · 2025-01-18
>
> # Response to Reviewer bUFZ
> We thank Reviewer bUFZ for providing a detailed valuable review and feedback. We have submitted a revised manuscript addressing many of the questions and suggestions raised during the discussion phase. The content in all appendix sections is newly added, and all modifications in the main text are highlighted in red.
>
> - A comparison of time complexity is required.
>
> In **Appendix E.3**, we provide detailed computational resources and a comprehensive comparison figure of model efficiency. Overall, while LSTI's computational overhead is moderate, its substantial improvement in accuracy far surpasses that of other methods, making it acceptable.
>
> - Request to include a more diverse range of datasets.
>
> We conduct additional experiments on two widely used datasets, Weather and ETTm1, with the results provided in **Appendix F**. The experiments reveal that LSTI achieves more than a 50% improvement on both datasets, highlighting its superiority in addressing long consecutive missing patterns.
>
> - Recommend providing detailed descriptions of each module.
>
> We include a detailed description of the Short-Term Imputer and the Meta-Learning Module in **Appendix C** and **Appendix D**.
>
> - Discussion on the Impact of LSTI Hyperparameters
>
> LSTI does not introduce any additional hyperparameters, which eliminates the need for hyperparameter tuning and ensures strong robustness.
>
> - Conduct more comprehensive ablation studies.
>
> We conduct ablation studies on all the loss functions employed in the Long-Term Imputer. Detailed results and analyses are provided in **Appendix B**.
>
> - Incorporate a broader impact
>
> We incorporate a description of the broader impact in **Appendix I**, covering areas such as healthcare, finance, and meteorology.
>
>
> - Provide the source code.
>
> We provide the training code for the Long-Term Imputer [here](https://anonymous.4open.science/r/LSTI-F03F). The full code will be available upon the publication of the paper.

---

> ### Author Response · Authors · 2025-02-06
> **Kind reminder regarding your feedback**
>
> Dear Reviewer bUFZ,
>
> We hope this message finds you well. We understand this might be a particularly busy time for you, and we genuinely appreciate the time you dedicate to reviewing. All the requested changes have been incorporated into the revised version, and further details can be found in our rebuttal.
>
> As we approach the final stages of the review process, we would kindly remind you to review our rebuttal at your earliest convenience. Your feedback is extremely valuable to us, and we would greatly appreciate any additional insights you could provide to help refine and strengthen our work. Should any aspects require further clarification, please do not hesitate to reach out.
>
> Thank you once again for your time, effort, and invaluable contribution to the review process. We eagerly look forward to your feedback.
>
> Best regards,
>
> Authors

---

### Comment · Reviewer_bUFZ · 2024-12-21
**Novel method for effectively handling consecutive missing values in time series data by combining long-term and short-term imputation strategies with a meta-weighting network**

This work presents a significant advancement in the field of time series imputation through the introduction of the Long Short-Term Imputer (LSTI), which effectively addresses the prevalent issue of consecutive missing values. The methodology is meticulously crafted, integrating both long-term and short-term imputation strategies, marking a substantial improvement over traditional methods that often focus on a singular aspect of temporal dependency. The authors' innovative use of bi-directional autoregression to capture long-term dependencies is particularly noteworthy, as it allows the model to leverage information from both past and future data points, thereby enhancing the robustness of the imputation process. This approach resonates with the findings of Li et al. [1] in their work, where the importance of capturing long-range dependencies in time series forecasting is emphasized.

The self-mapping network designed for short-term dependencies is another commendable aspect of this work, showcasing a sophisticated understanding of the diverse nature of missing data patterns. This is crucial for effective imputation, especially in scenarios where short-term fluctuations can significantly impact the overall data integrity. The introduction of the meta-weighting module is a particularly strong feature, as it provides an adaptive mechanism that intelligently balances the contributions of the long-term and short-term imputers based on the specific characteristics of the input data. This innovation not only enhances the model's flexibility but also its overall performance, as evidenced by the substantial reduction in error rates compared to existing SOTA methods, such as those proposed by [2], which often struggle with long-interval consecutive missing values.

The extensive experiments conducted across multiple real-world datasets lend significant credibility to the findings and highlight the practical applicability of the proposed approach. However, the paper could be further strengthened by providing a more detailed discussion on the computational complexity and scalability of the LSTI model. As time series datasets continue to grow in size and complexity, understanding the model's performance in such scenarios is critical for its adoption in real-world applications. Additionally, while the results are impressive, a deeper exploration of the limitations of the model, particularly in handling extreme cases of missing data or highly irregular time series, would provide a more balanced perspective.

Moreover, the paper could benefit from a more comprehensive comparison with a broader range of existing imputation techniques, including those that utilize different underlying architectures, such as convolutional neural networks or hybrid models. For instance, comparing the LSTI model with the architecture proposed by [3] could provide valuable insights into its relative strengths and weaknesses. Furthermore, a discussion on how the LSTI model can be adapted or extended to handle non-stationary time series data, as explored by [4] would enhance the paper's relevance and applicability [4].

[1] "Time Series Forecasting With Deep Learning: A Survey" - Lim et al.

[2] "Deep Learning For Time Series Forecasting: Tutorial and Literature Survey" - Benidis et al.

[3] "A Comprehensive Survey of Time Series Forecasting: Architectural Diversity and Open Challenges" - Kim et al.

[4] "Deep Learning Models for Time Series Forecasting: A Review" - Li et al.

---

### Author Response · Authors · 2025-01-18

# General comments

We would like to express our sincere gratitude to all the reviewers for their valuable time and effort in reviewing our paper. In this general response, we summarize the key revisions made in the updated manuscript and aim to address each of the specific points raised in the individual rebuttals. The content in all appendix sections is newly added, and all modifications in the main text are highlighted in red. The main changes include:

- In Section 3, we revise the continuous missing data formulation by removing the Q matrix, simplifying the expression.

- In Section 4, we add a figure illustrating the Blackout missing pattern, along with a enhanced description.

- In Appendix A, we present LSTI's variability and confidence estimates, including the standard deviation, confidence intervals, and cumulative average convergence.

- In Appendix B, we provide a detailed description of the loss functions used in the Long-term Imputer and include an additional ablation experiment to demonstrate their effectiveness.

- In Appendices C and D, we offer detailed supplementary descriptions of the Short-term Imputer and the Meta-Learning Module, respectively.

- In Appendix E, we provide further details on the experimental setup and datasets, along with a visual showcase and a comprehensive comparison of model efficiency across various datasets.

- In Appendix F, we  present additional experiments conducted on the widely used Weather and ETTm1 datasets. The results demonstrate that LSTI achieves an improvement of approximately 50% on both datasets.

- In Appendix G, we provide additional comparative experiments with the SSSD method and provide a detailed analysis of why SSSD underperforms relative to LSTI.

- In Appendix H, we provide an analysis of the strengths and limitations of INR-based methods in the imputation domain and present additional comparative experiments, using TimeFlow as a representative method.

- In Appendix I, we provide a Broader Impact analysis across various domains, including healthcare, finance, and environmental science.

---

### Decision · Action_Editor_1BWW · 2025-02-06

**Recommendation:** Accept with minor revision

**Comment:**

All reviewers are leaning towards accepting the paper. The authors fully respond to the majority of the reviewers' remarks, the paper improved after the revision, especially it is much clearer to read and the experimental results are good.

Therefore I accept (with minor revision) the paper for publication in the TMLR.

However, the following concerns listed by Reviewer 5gmT remain, and I encourage the authors to incorporate appropriate changes in the final revision of their paper:
"The improvements compared to all other considered baselines are impressive but sometimes questionable. All the considered datasets exhibit clear patterns and frequencies, and I do not understand why some results for strong baselines are this poor. There may be aspects of the training procedure that I do not fully understand. The paper still lacks qualitative comparisons, such as more plots and a specific focus on a dataset in the appendix."

**Audience:**

The paper falls into the area of interest of the TMLR's audience working on machine learning methods for time series.

**Claims And Evidence:**

The reviewers agreed that the claims are supported by evidence being significant enough.

---

> ### Author Response · Authors · 2025-02-24
> **Camera-ready manuscript available**
>
> Dear AE,
>
> We have uploaded the camera-ready version of our manuscript.  In response to your decision letter, we have made the following revisions:
>
> First, in the previous version, we followed the dataset splitting strategy from the original TimeFlow paper, which led to suboptimal experimental results. In the camera-ready version, we carefully tuned the TimeFlow in experiments, resulting in a substantial improvement in performance, though it required significantly more time and memory. The updated results can be found in Tables 15 and 16. Overall, under the Blackout missing pattern, TimeFlow incurs a computational cost roughly 4-5 times higher than LSTI, yet LSTI continues to outperform TimeFlow.
>
> Second, we added Table 10 to provide detailed information about the newly added datasets in the appendix.
>
> Once again, we sincerely appreciate your time, effort, and valuable feedback throughout the review and submission process. We are confident that your input has significantly improved our work.
>
> Best regards,
>
> Authors